# Transcriptional responses of cancer cells to heat shock-inducing stimuli involve amplification of robust HSF1 binding

Sayantani Ghosh Dastidar[1,2], Bony De Kumar[1,3], Bo Lauckner[1], Damien Parrello[1], Danielle Perley[1,4], Maria Vlasenok [1,5], Antariksh Tyagi[1,3], Nii Koney-Kwaku Koney [1,6], Ata Abbas [1,7] & Sergei Nechaev [1] ✉

Responses of cells to stimuli are increasingly discovered to involve the binding of sequence-specific transcription factors outside of known target genes. We wanted to determine to what extent the genome-wide binding and function of a transcription factor are shaped by the cell type versus the stimulus. To do so, we induced the Heat Shock Response pathway in two different cancer cell lines with two different stimuli and related the binding of its master regulator HSF1 to nascent RNA and chromatin accessibility. Here, we show that HSF1 binding patterns retain their identity between basal conditions and under different magnitudes of activation, so that common HSF1 binding is globally associated with distinct transcription outcomes. HSF1-induced increase in DNA accessibility was modest in scale, but occurred predominantly at remote genomic sites. Apart from regulating transcription at existing elements including promoters and enhancers, HSF1 binding amplified during responses to stimuli may engage inactive chromatin.

Transcriptional responses to stimuli involve the binding of sequence-specific factors to DNA to regulate their target genes. The Heat Shock Response (HSR) is an evolutionarily conserved cellular stress defense mechanism that can be triggered by heat and some other stimuli[1–6], and is frequently active in cancers[4,7]. Even though HSR is recognized to be a complex pathway involving numerous regulatory components[8–11], the Heat Shock Factor (HSF, HSF1 in mammals) is considered its master regulator. HSF1 is normally found in the cytoplasm[3,12], but during stress is phosphorylated to translocate into the nucleus and bind to its cognate Heat Shock Response DNA Elements (HREs)[4,13,14]. Foundational work in *Drosophila* demonstrated that HSR involves rapid appropriation of the RNA polymerase II (Pol II) machinery by a handful of Heat Shock Protein (*HSP*) genes, leading to their massive activation at the expense of the rest of the genome[15,16]. Binding to promoters of *HSP*

genes to activate their transcription was presumed to be the function of HSF[17–19]. This paradigm has shaped our understanding of transcription and roles of DNA sequence-specific transcription factors in regulation[12,20,21]. However, recent studies revealed that the binding of HSF1 during HSR is not limited to *HSP* gene promoters, or to promoters at all, but also occurs at thousands of intergenic loci[8,22]. Moreover, HSF1 binding near promoters does not necessarily lead to transcription activation of nearby genes[2,8]. These findings challenged the established relationship between the binding of a transcription factor to its target sites and its function in transcription.

Analysis of the human genome identifies over two hundred and eighty thousand HSF1 binding motifs in the hg19 assembly[23]. In contrast, profiling using Chromatin-Immunoprecipitation Sequencing (ChIP-seq) in human[4,22,24] or mouse cells[6,8] finds only on the order of ten

[1]Department of Biomedical Sciences, University of North Dakota School of Medicine, Grand Forks, ND 58202, USA. [2]Illumina, Inc., San Diego, CA 92122, USA. [3]Yale Center for Genome Analysis, Department of Genetics, Yale University School of Medicine, New Haven, CT 06510, USA. [4]Canadian Centre for Computational Genomics, McGill Genome Centre, Montreal, QC H3A0G1, Canada. [5]Center for Molecular and Cellular Biology, Skolkovo Institute of Science and Technology, Moscow 121205, Russia. [6]University of Ghana Medical School, University of Ghana, Accra, Ghana. [7]Department of Biochemistry, Case Western Reserve University, Cleveland, OH 44106, USA. ✉e-mail: sergei.nechaev@UND.edu

thousand peaks per dataset[25], leaving ample room for transcription factor binding patterns to be flexible. Earlier work reported different HSF1 programs between steady-state cancer cells and noncancerous cells responding to heat[4]. Heat shock in cancer cells was also shown to induce widespread HSF1 binding[22]. However, experimental information on transcription factor binding between cell types or in response to different stimuli remains limited. We therefore performed a side-by-side comparison of HSR induced with different stimuli in distinct ground states. To do so, we subjected distant human cell lines, MCF-7 breast adenocarcinoma and K562 chronic myelogenous leukemia, to two different HSR-inducing stimuli, elevated temperature and arsenic, a toxic metalloid found in soil, air and water, at the ambient temperature[26].

Here, we show that global HSF1 binding patterns retain cell type specificity under different stimuli and magnitudes of activation. By following the binding of HSF1 along with nascent RNA and chromatin accessibility, we show that HSF1 favors pre-existing open chromatin sites across the genome and that its global binding patterns are more stable than transcription. Apart from context-specific transcription activation of promoters and enhancers, stimulus-induced HSF1 binding may engage inactive chromatin sites.

## Results

### HSF1 binding is a temperature-independent hallmark of HSR

We began by examining MCF7 breast adenocarcinoma cells during temperature-induced Heat Shock Response (HSR) (Fig. 1, Supplementary Data 1, Supplementary Fig. 1a). A 60-min incubation at 42 °C induced a control HSP gene seen as spreading of the Pol II and nascent RNA signal into the gene body (Fig. 1a). HSF1 showed characteristic binding at the *HSPH1* gene promoter near the transcription start site (TSS) (Fig. 1a), similar to that widely observed on HSP genes[4,8]. Genome-wide, ChIP-sequencing using a previously validated anti-HSF1 antibody[22] showed widespread HS-dependent signal at thousands of sites at (+/−1 kb) and outside of annotated gene promoters (Fig. 1b, c), with ~18,000 peaks (*q* < 0.01) identified between independent biological replicates in HS, compared to up to ~2400 peaks before activation

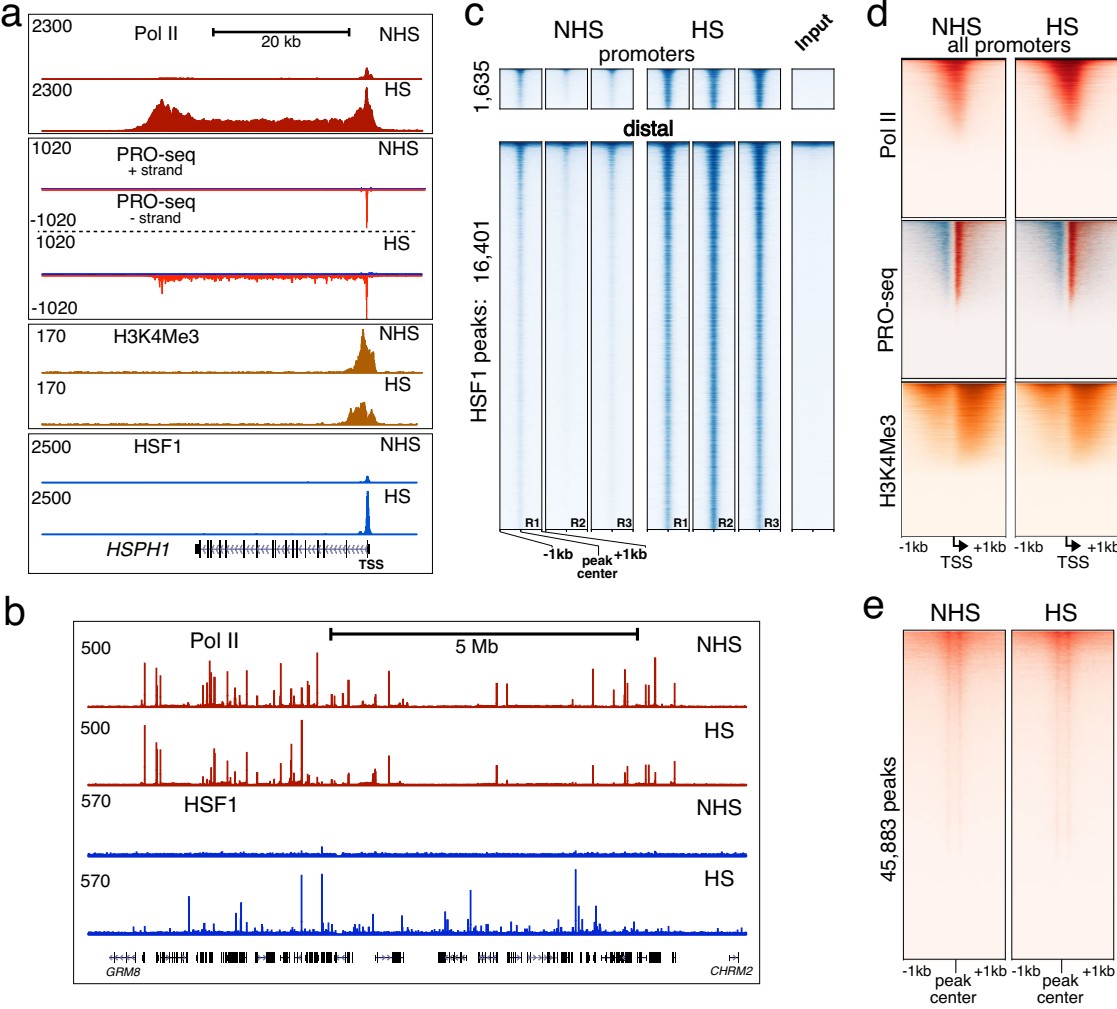

**Fig. 1 | Genome-wide response to HS. a** UCSC genome browser tracks of ChIP-seq and PRO-seq datasets for MCF7 cells in non-heat shock (NHS) and heat shock (HS) conditions are shown on a HS-inducible gene *HSPH1*. PRO-seq tracks are separated by genomic plus and minus DNA strands. The gene transcription start site (TSS) is indicated. **b** Pol II and HSF1 ChIP-seq tracks for NHS and HS conditions showing a randomly selected section of the genome (hg19, chr7:126M-137M) containing ~75 unique genes. **c** Heatmaps showing HSF1 ChIP-seq signal centered at HSF1 peaks in promoter-proximal and distal regions in NHS and HS conditions, sorted by peak intensity in HS replicate 2 and shown for all individual replicates as well as a ChIP input sample. **d** Heatmaps showing the indicated readouts centered at TSSs for all 23 K promoters. PRO-seq signal is shown in red for the sense strand signal and in blue for signal antisense with respect to genes. **e** Heatmap of PRO-seq signal centered around PRO-seq peaks located outside of gene regions with respect to the plus strand of the genome. The data indicate prevalent pausing at positions on either side approximately 50-nt from the peak center. Heatmaps in (**d**) and (**e**) are sorted by signal intensity in HS samples and all datasets are scaled to their sequencing depths.

(Supplementary Fig. 1b; Supplementary Data 2). A different anti-HSF1 antibody[4] showed 75% overlap in peak locations in MCF7 cells (Supplementary Fig. 1c, d). Genome-wide changes for other readouts, however, including transcription itself, were less pronounced. An active promoter histone mark H3K4me3 showed no changes in HS compared to untreated control (non-heat shock, NHS) cells (Fig. 1a, d, Supplementary Fig. 1e). There was even a modest decrease in H3K4me3 signal at the immediate promoter-proximal regions of highly activated genes including *HSP70 (HSPA1B)* (Fig. 1a, Supplementary Fig. 1e), similar to findings in human embryonic stem cells[24] (Supplementary Fig. 1f) and *Drosophila*[27], and consistent with earlier reports on nucleosome dynamics[28–30]. Nascent RNA analysis using Precision nuclear Run-On sequencing (PRO-seq) showed 2288 genes upregulated in HS (p-adj < 0.05) (Supplementary Data 3). Pol II ChIP-seq and PRO-seq signal was retained at transcription initiation sites both at and outside of promoters (Fig. 1a, d, e, Supplementary Fig. 1g), consistent with ongoing transcription in HS outside of activated genes genome-wide[8,31].

Upregulation of *HSP* genes has been previously induced at the ambient temperature in the presence of inorganic arsenic[26]. This prompted us to ask whether arsenic would induce genome-wide HSF1 binding at the ambient temperature in these cells. Treatment of MCF-7 cells with sodium meta-arsenite (As) at 37 °C showed transcriptional activation of control HS genes, with the timing and magnitude of mRNA level increase resembling those induced by heat (Fig. 2a, Supplementary Fig. 2a). Like HS, As induced characteristic HSF1 and Pol II binding at *HSPA1B* gene promoter as observed by ChIP-qPCR (Fig. 2b). ChIP-sequencing of As-treated MCF7 cells showed widespread HSF1 binding (Fig. 2c, d, Supplementary Data 2). HSF1 signal in each treatment at promoters was comparable in intensity, although statistically higher than at distant regions (p < 0.0001) (Fig. 2e), with the intensity of HSF1 signal being comparable between HS and As treatments (Supplementary Fig. 2b). Elevated temperature is therefore not a prerequisite for widespread genome-wide HSF1 binding in cancer cells.

Comparing HSF1 binding locations between HS and As treatments, the majority of peaks in MCF7 cells overlapped between them (Fig. 2f). Accordingly, sorting all HSF1 peaks by the signal intensity in one treatment retained their ordering in the other (Fig. 2d). All groups of peaks were dominated by the cognate HRE DNA sequence motifs (Supplementary Fig. 2c). Examining nascent transcription in PRO-seq datasets, As showed robust activation (2974 genes, p-adj < 0.05, Supplementary Data 3). Genes commonly activated in HS and As were

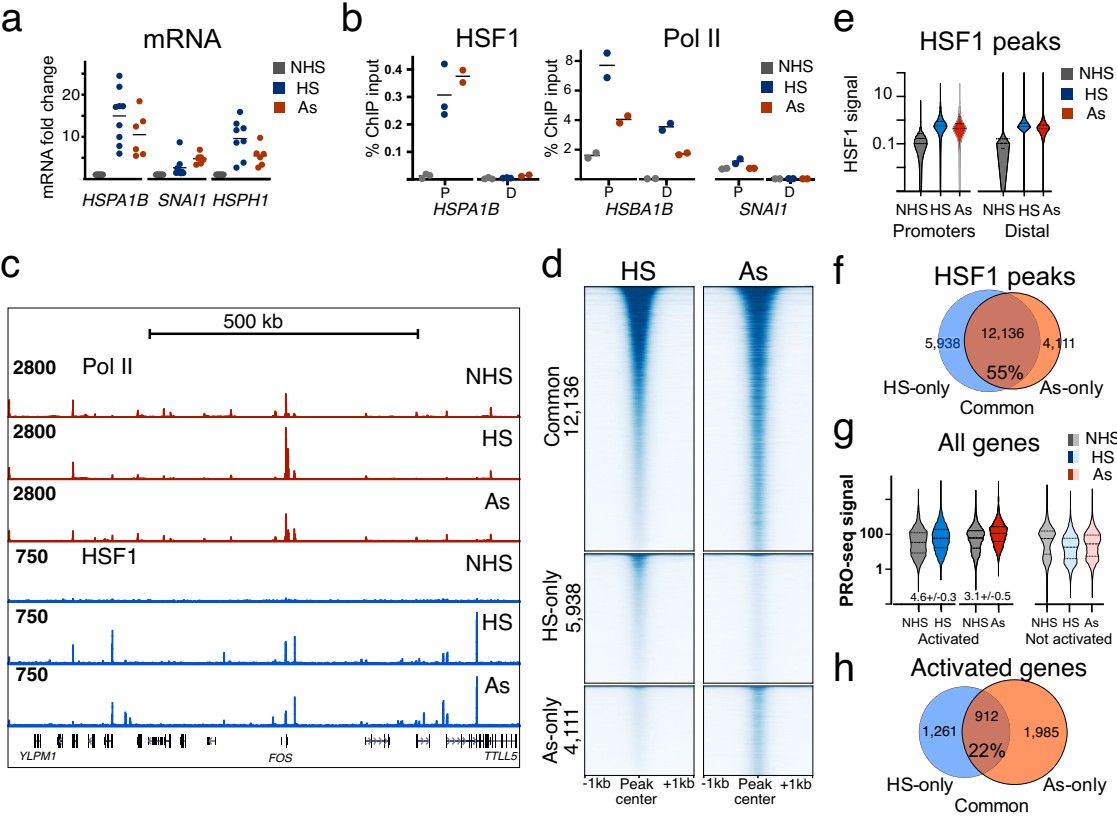

**Fig. 2 | Widespread HSF1 binding is induced by As treatment. a** RT-qPCR showing fold change for the indicated mRNA transcripts normalized to *GAPDH* in NHS, HS, and As conditions. This information was derived from 9 independent biological replicates (HS) and 6 replicates (As), which are shown individually. The line shows the mean. **b** ChIP-qPCR against HSF1 and RNA Pol II showing fold enrichment over input at promoter and distal regions of *HSPA1B* and *SNAI1* genes in NHS, HS, and As conditions. This information was derived from 3 independent biological replicates for HSF1 and 2 replicates for Pol II. Dots in (**a**) and (**b**) show independent biological replicates. **c** UCSC genome browser tracks showing ChIP-seq signal from RNA Pol II and HSF1 on a ~1 Mb genomic region (hg19, chr14: 75.2 M–76.2 M) surrounding *FOS* gene that is activated in HS. **d** Heat-map of peak-centered MCF7 HSF1 signal in HS and As conditions for peaks common for HS and As, exclusive for HS and exclusive for As. **e** Averaged HSF1 signal (Counts Per Million uniquely mapped reads) from peaks at the promoter (TSS +/−1000) or distal regions at the indicated conditions. Promoter signal was higher than intergenic signal for As (p < 0.001) and HS replicates (p < 0.045, p < 0.0001) based on the Mann–Whitney test. **f** Venn diagram showing the numbers of common and exclusive MCF7 HSF1 peaks between HS and As conditions. The percentage of common peaks among all peaks is shown. **g** PRO-seq gene body signal density for activated and not activated genes in MCF7 cells. Data for this graph were normalized by the sequencing coverage after rRNA removal. Numbers indicate the mean fold activation compared to the same genes in untreated cells, relative to not activated genes, with the range between two biological replicates. PRO-seq counts are log-transformed for violin plots. **h** Venn diagram showing common and treatment-exclusive activated genes. The percentage of commonly activated genes is shown. Source data are provided as a Source Data file.

enriched in stimulus-response categories including classic *HSP* genes (Supplementary Data 4). Genes activated only in HS did not fall into specific categories, while genes activated only in As, based on fold-overrepresentation, were enriched for responses to metal ions (Supplementary Data 4). The fold gene activation was modestly higher in HS than As treatments in our hands (Fig. 2g), with As-induced transcription reaching saturation in titration experiments (Supplementary Fig. 2d). Comparing HSF1 binding and gene activation, HSF1 peaks showed a higher overlap between treatments than did transcriptionally activated genes (Fig. 2f, h), with only about a fifth of activated genes being in common between treatments. This persisted under increased stringency of calling HSF1 peaks or activated genes (Supplementary Fig. 2e). Taken together, widespread HSF1 binding is a temperature-independent hallmark of HSR whose genome-wide patterns are more stable between stimuli than transcription outcomes.

## Common HSF1 binding and variable transcription activation

Two treatments inducing widespread HSF1 binding allowed us to probe the relationship between HSF1 promoter binding and gene transcription. Approximately 8% of all gene promoters contained HSF1 peaks (Fig. 3a). For activated genes, the fraction of promoters with HSF1 peaks was at least 2-fold higher (Fig. 3b), broadly implicating HSF1 binding in gene activation. Known HSP genes showed HSF1 binding and transcription activation similar to previous reports in human and mouse cells[2,8,22], with most of these genes activated in both

treatments (Supplementary Fig. 3a). Outside of HSP genes, however, a majority of genes were activated in the absence of HSF1 binding (Fig. 3b) and, accordingly, most HSF1 binding at promoters was not associated with gene activation (Fig. 3c). These observations are consistent with previous work in HS[2,8] and reinforce the notion of the overall disconnect between HSF1 binding and nearby gene transcription. To gain insight into possible reasons behind this disconnect, we compared HSF1 binding on activated versus not activated genes. While HSF1 peaks did show higher HSF1 signal at promoters of activated genes (Fig. 3d), transcription fold-activation was similar for genes activated with and without HSF1 (Fig. 3e). Promoters containing HSF1 showed a higher overlap between treatments regardless of gene activation status (Supplementary Fig. 3c, d). Surprisingly, the difference between HSF1-bound and unbound promoters was in basal transcription in NHS cells, which was higher for genes whose promoters would bind HSF1 in HSR than those that would not. This held true for activated (Fig. 3e) and not activated genes alike (Fig. 3f). HSF1 thus favors basally active promoters regardless of whether the genes are activated in HSR or not.

We next examined a subset of ~900 activated genes that showed HSF1 peaks at their promoters. A vast majority of HSF1 peaks at promoters of genes activated exclusively in one treatment were common between treatments (Fig. 3g). HSF1 peaks unique to each treatment were relatively few, but were enriched in genes activated in the same treatment (HS-only and As-only, Fig. 3g). Pivoting the data to view

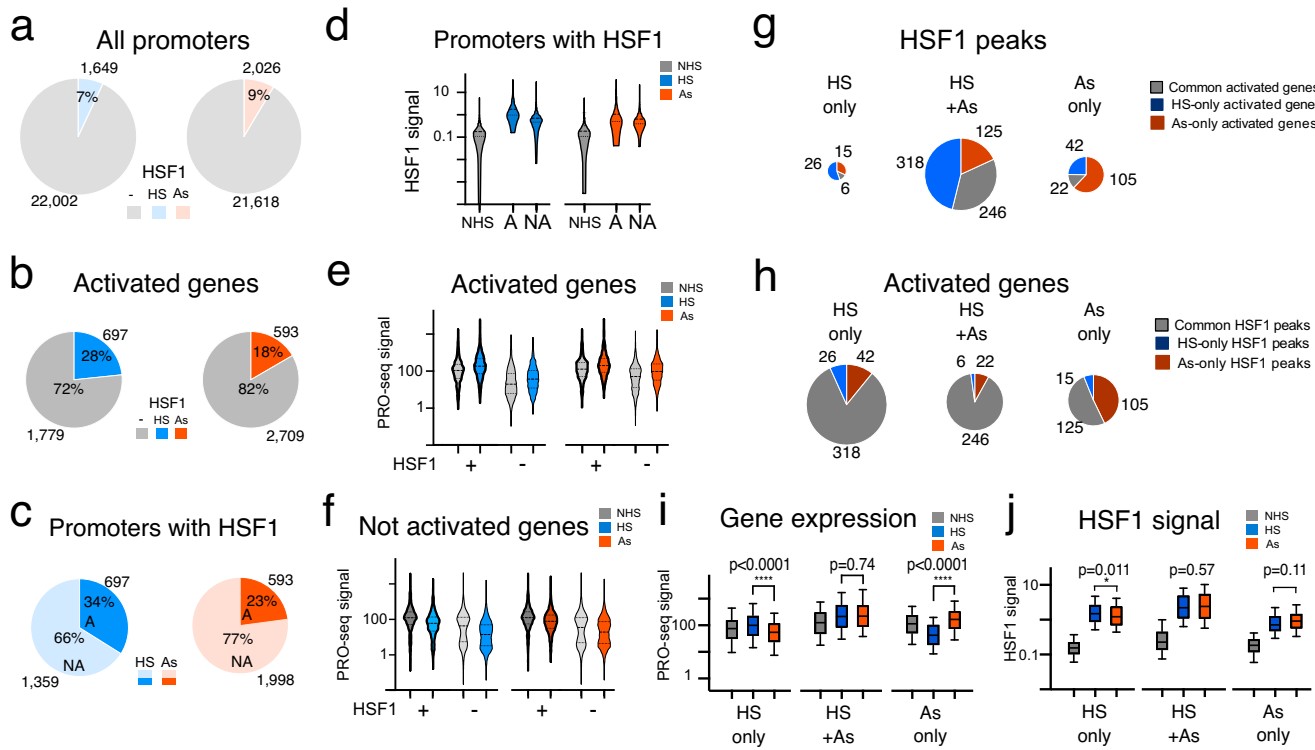

**Fig. 3 | HSF1 promoter binding and gene activation.** Pie charts show the numbers and fractions of promoters with or without HSF1 peaks within +/1 kb interval for **a** all 23 K promoters and **b** promoters of activated genes. **c** Pie charts show the numbers and proportion of promoters containing HSF1 within +/−1 kb from the TSS for genes that are transcriptionally activated in the same treatment (darker color) or not activated (lighter color). **d** HSF1 peak signal for genes containing HSF1 peaks at promoters as in (**c**) for activated (A) and not activated (NA) genes, shown in a truncated violin plot. **e, f** Gene expression signal based on PRO-seq gene body count density for genes with (+) and without (-) HSF1 peaks at promoter regions (+/−1 kb from the gene TSS). Shown are activated genes (**e**) and not activated genes (**f**). PRO-seq counts are log-transformed for violin plots. **g** HSF1 binding and gene activation between treatments. Common and treatment-unique HSF1 peak

categories are indicated in pie charts scaled to the number of genes, with slices indicating common (grey) or treatment-unique activated genes (blue for HS and red for As). **h** Distribution of HSF1 peaks among genes activated in both (HS+As) treatments or uniquely in each treatment. Slices indicate common (grey) or treatment-unique HSF1 peaks at promoters. **i** Gene expression based on PRO-seq read density for gene groups in (**h**) for HS only (386 genes), HS + AS common (274 genes) and As only genes (245 genes). **j** HSF1 peak signal at NHS, HS, and As conditions for the same genes shown in (**h**) and (**i**). Box plots show medians, upper and lower quartiles, with whiskers indicating 10-90 percentiles. Statistical significance is calculated based on two-tailed *p*-value using Mann–Whitney test. Source data are provided as a Source Data file.

activated genes, we noted no enrichment of HS-specific HSF1 peaks at genes activated in HS (Fig. 3h). We did note that 43% of As-exclusive activated genes were associated with As-only HSF1 promoter peaks (Fig. 3h), indicating that HSF1 can bind and potentially function at new sites outside of a generic response to temperature. However, As-exclusive activation involved a modest number of genes, and a vast majority of activated genes were associated with common HSF1 binding (Fig. 3h). Despite the differences in transcriptional outcomes, HSF1-bound genes activated exclusively in HS or in As did not show significant differences in HSF1 binding intensity by stringent criteria ($p = 0.011$) (Fig. 3i, j) and no difference in binding positions (Supplementary Fig. 3e). Taken together, comparison of two HSR-inducing stimuli points to widespread decoupling of HSF1 binding and gene activation, with context-specific transcription predominantly associated with common HSF1 binding.

## Distinct HSF1 patterns between cell lines

Having compared HSR in MCF7 cells induced with HS or As, we applied the same two stimuli to cells of an unrelated origin. K562 is a Tier I ENCODE leukemia cell line that has been previously examined for rapid responses to heat[22]. Both MCF7 and K562 cells responded to HS or As by upregulating control genes (Figs. 4a, 2a). Genome-wide, HS

treatment of K562 and MCF7 cells activated over a thousand genes (1760 and 2288, respectively) (Supplementary Data 4), with 70% of upregulated genes (546 out of 778 genes) from an existing K562 HS dataset, including all HSP genes, overlapping with our K562 HS treatment[31] (Supplementary Fig. 3b). HS and As induced HSF1 activation (Fig. 4b) and widespread DNA binding in both cell lines (Fig. 4c, d, Supplementary Data 2), indicating that temperature independence of genome-wide HSF1 binding is not confined to a particular cell type.

The overlap in HSF1 peak locations between the two cell lines was lower than that between treatments (Fig. 4e, Fig. 2), consistent with cell type specificity of HSF1 binding. However, the overlaps between HSF1 peaks were still higher than between activated genes (Fig. 4e, f), consistent with higher conservation of HSF1 binding than transcriptional responses. Comparing HSF1 signal across all datapoints showed clear rank-ordering by the cell type (Fig. 4d) and more extensive changes between cell types than between treatments (Supplementary Fig. 4a, b). By Principal Component Analysis (PCA), individual datasets also separated by the cell type (Fig. 4g)[32]. This separation was preserved under different magnitudes of HSR activation (Supplementary Fig. 4c) and when a subset of high confidence common peaks was used (Supplementary Fig. 4d). Differences in HSF1 binding intensity can thus account for cell type-specific binding patterns.

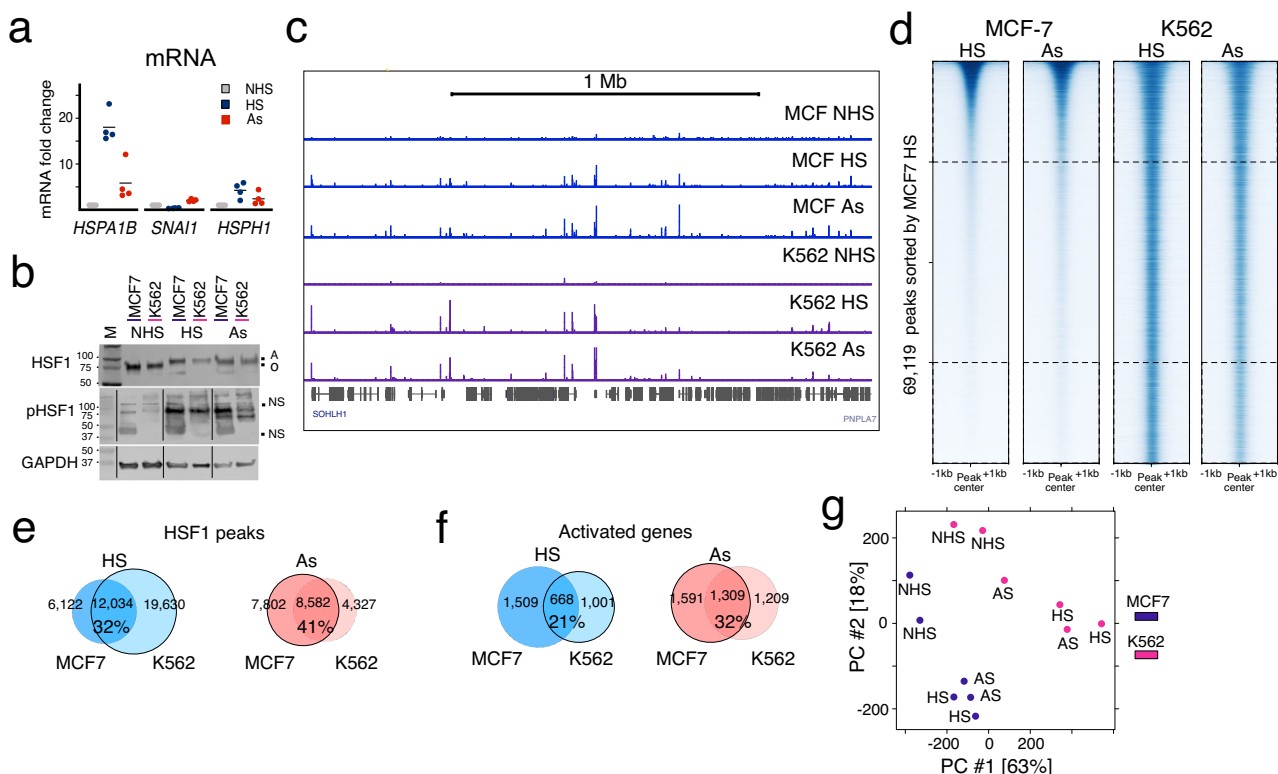

**Fig. 4 | Cell type specificity of genome-wide HSF1 binding under different stimuli. a** RT-qPCR showing fold-change of indicated mRNAs compared to *GAPDH* in K562 cells. This information was derived from 4 independent biological replicates. **b** Western Blot showing pan HSF1 and phosphorylated HSF1 (pHSF1) in indicated conditions in MCF7 and K562 cells. GAPDH is used as a loading control. NS represents non-specific bands. A and O represent active (phosphorylated) and inactive forms of HSF1 distinguished by different mobility on SDS gels. Numbers indicate marker band molecular sizes in kDa. Images for the GAPDH and pHSF1 blot, provided in the Source Data file, were digitally shifted to remove empty wells as indicated by vertical black lanes. Pan HSF1 and pHSF1 blots were run on separate gels. This experiment has been repeated twice with similar results. **c** UCSC browser shot showing MCF7 (blue) and K562 (purple) HSF1 ChIP-seq tracks from a randomly selected portion of the genome (chr9: 138.5 M–140.5 M) containing ~100 uniquely named genes. **d** HSF1 peak signal (+/−1 kb from peak center) in MCF7 and K562 cells shown in heatmaps sorted by MCF7 HS signal. Dashed lines indicate top and bottom quartiles. **e** Venn diagrams showing the overlap between HSF1 peaks in HS (blue) and As (red) treatments between MCF7 and K562 cells. The percentage of common peaks among all peaks is shown. **f** Venn diagram showing activated genes in HS (blue) and As (red) conditions in MCF7 and K562 cells. **g** PCA plot of HSF1 peaks in MCF7 (purple) versus K562 (pink) cells shown for each condition per replicate. Source data are provided as a Source Data file.

## Global connection between HSF1 binding and nascent transcription

Even though over 80% of all HSF1 peaks were found within intergenic regions, their density was several-fold higher at gene promoters despite the lack of enrichment of HRE sequence elements (Fig. 5a). Given that most HSF1 binding at promoters did not activate transcription (Fig. 3c)[8], we asked to what extent HSF1 binding associated with gene activation is conserved between distant cell lines. HSF1 showed higher enrichment at promoters of genes activated in both cell lines than at those activated exclusively in one, with about half of promoters for genes commonly activated between the cell lines containing HSF1 peaks (Fig. 5b). These commonly activated genes included the known heat shock response genes harboring peaks with the highest signal (Supplementary Data 5). However, this group makes only a modest fraction of genes activated between these two cell lines (Figs. 5b, 4f, Supplementary Fig. 3a), so that the strong connection between HSF1 binding and transcription activation on these genes is effectively diluted when viewed genome-wide (Fig. 5b, c).

Outside of promoters, HS was shown to involve intergenic sites previously referred to as distant Transcription Regulatory Elements (dTREs)[31]. We identified dTREs in our HS PRO-seq datasets based on nascent RNA signatures (Supplementary Data 6, Supplementary Fig. 5a). We do not refer to these elements as enhancers, although the identified dTREs overlapped with about 50% of enhancers previously described in either cell line[33]. The proportion of dTREs with HSF1 peaks was also higher among dTREs unique to HS compared to NHS conditions (Supplementary Fig. 5b). Examining DNA sequences, we noted an enrichment of HSF1 motifs around HS-induced dTREs (Supplementary Fig. 5c). However, the density of HSF1 peaks at dTREs was higher than the genomic density of HRE sequence motifs (Fig. 5d), indicating that, similar to promoters, HSF1 is recruited to dTRE sites by means other than HRE sequences alone. Like promoters, most dTREs were activated in the absence of HSF1 binding (Supplementary Fig. 5b). A group of HS-activated dTREs common between the two cell lines showed the enrichment of HSF1 peaks that was at least as high as that for promoters of commonly activated genes (Fig. 5b, e). Gene Ontology (GO) analysis for genes associated with these dTREs that were previously annotated as enhancers[33] did not show statistically significant gene categories, but did include HS-induced heat shock protein *HSPA9O* and *HSPA8* genes. However, this group of conserved dTREs was small (Fig. 5e) and, similar to promoters, it did not sway the overall disconnect between HSF1 binding and transcription (Fig. 5c, f; Supplementary Fig. 5d). The proportion of all dTREs containing HSF1 peaks was similar to that at promoters as well (Fig. 5c, f), reinforcing a modest connection between HSF1 binding and transcription activation as a genome-wide property of HSR.

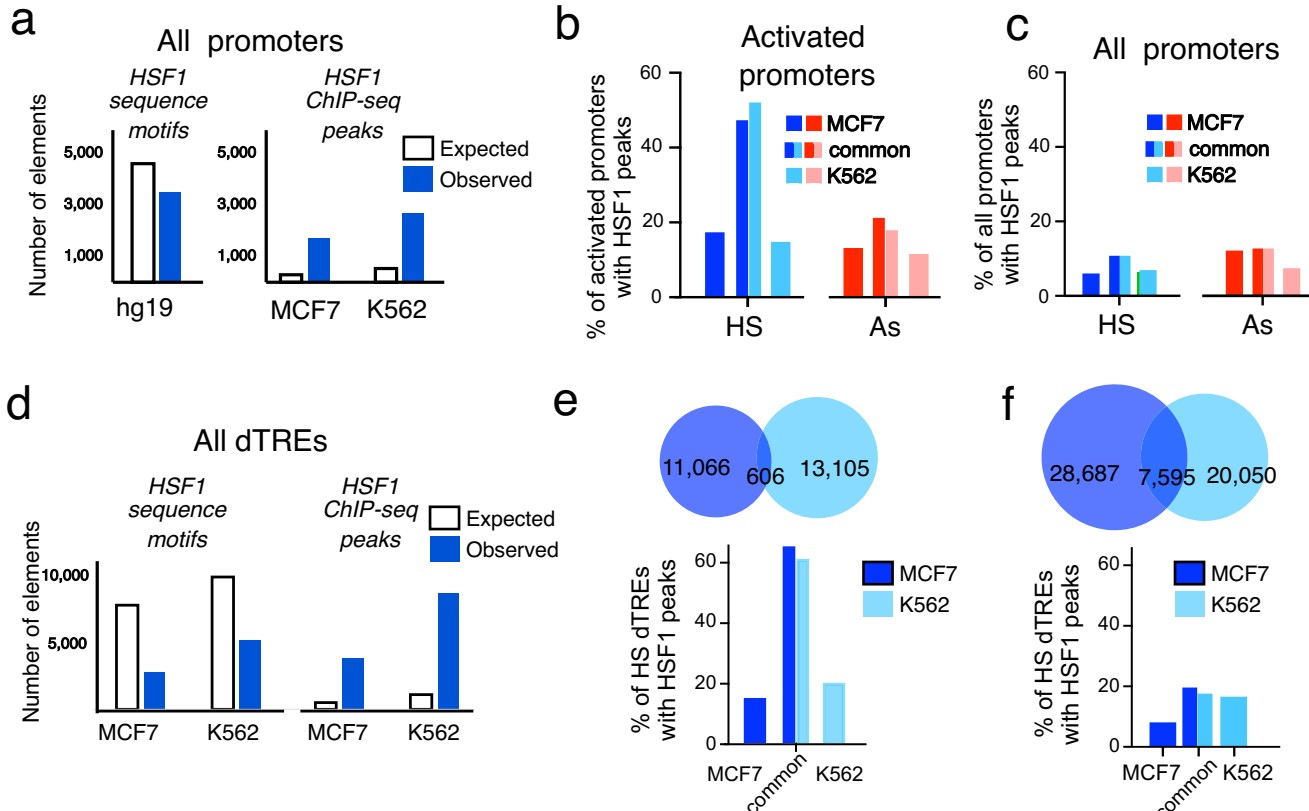

**Fig. 5 | Genome-wide disconnect between HSF1 binding and transcriptional activation. a** The numbers of observed HSF1 sequence motifs and ChIP-seq peaks in the vicinity of promoter regions (blue) compared with motifs and peaks in these regions expected at random (white). Observed motifs are based on homer's find-motif output for hg19 genome +/−1 kb from the TSSs. Expected motifs are calculated assuming even distribution of the same number of HSF1 sequence motifs (~286 K) and HSF1 peaks (~18 K) across the genome. **b** Fractions of activated promoters in HS (blue) and As (red) in MCF7 (dark blue, dark red) and K562 (light blue, light red) cells. Approximately 50% (in HS) and 20% (in As) of commonly activated genes showed HSF1 at their promoters. **c** Fractions of HSF1 peaks at all gene promoters in HS (blue) and As (red) in MCF7 (dark blue, dark red) and K562 (light blue, light red) cells, at promoters common to both or exclusive to each cell line. **d** The expected and observed numbers of HSF1 sequence motifs and ChIP-seq peaks are shown around dTREs as in (**a**). Unlike promoters, dTREs are cell line specific, so that HSF1 sequence motifs are shown separately for each cell line. **e** HS-activated dTREs between K562 and MCF7 cells. Venn diagram showing their overlap between the two cell lines. The percentages of dTREs with HSF1 peaks for dTREs common between the cell lines and exclusive dTREs for each cell line are shown underneath. **f** Same as in (**e**) except all dTREs are shown.

## HSR involves amplification of basal HSF1 binding

To understand how the genome-wide HSF1 binding may be established from the ground state, we compared HSF1 signal during HSR and at non-heat shock (NHS) conditions, considering two possibilities. First, HSF1 may occupy different sites before and during HSR. However, a vast majority of HSF1 peaks detected in NHS datasets were in common with HS samples (Fig. 6a, Supplementary Fig. 6a). A small number of peaks unique to NHS (Fig. 6a, Supplementary Data 7) were found mostly at promoters (42 out of 59 NHS-exclusive peaks in MCF7 cells compared to HS). The corresponding genes were metabolism-related and were not activated in HS (Supplementary Data 7). Manually examining high-level signal showed nonspecific pileups in all samples (Supplementary Fig. 6b, c), validating, if fortuitously, our sequencing depth-based normalization. NHS-specific events thus comprise at most a tiny proportion of HSF1 binding.

Another possibility is that HSF1 may bind to the same sites before and after HSR. In this case, low-intensity HSF1 signal should be evident in NHS cells at the locations of HSF1 peaks that would be newly

acquired in HSR. The average HSF1 signal in NHS cells at HS peak locations, including HS-only peaks, was above that at randomly selected background regions (Fig. 6b). Unlike the peak regions, HSF1 signal at these background regions did not increase in HS (Supplementary Fig. 6d). Examining individual loci, we see low-level specific signal in our NHS datasets (Supplementary Fig. 6e, f) and in previously published NHS data (Supplementary Fig. 6g). Amplification of low-level binding, therefore, appears to be the primary mode of HSR. This amplification can be either uniform across all sites or selective. To address, we compared HSF1 signal in NHS cells at the locations of HSR peaks arising exclusively in each cell line or in each treatment. NHS signal at the locations of cell line-exclusive peaks positively correlated with HSR signal in the same, but not the opposite cell line (Fig. 6c), consistent with cell line specificity of basal HSF1 binding. On the other hand, NHS signal at the locations of peaks exclusive to one treatment correlated with HSR signal in both treatments for each cell line (Fig. 6d, e), indicating that basal HSF1 binding anticipates responses to both stimuli and arguing for selectivity of signal amplification. However,

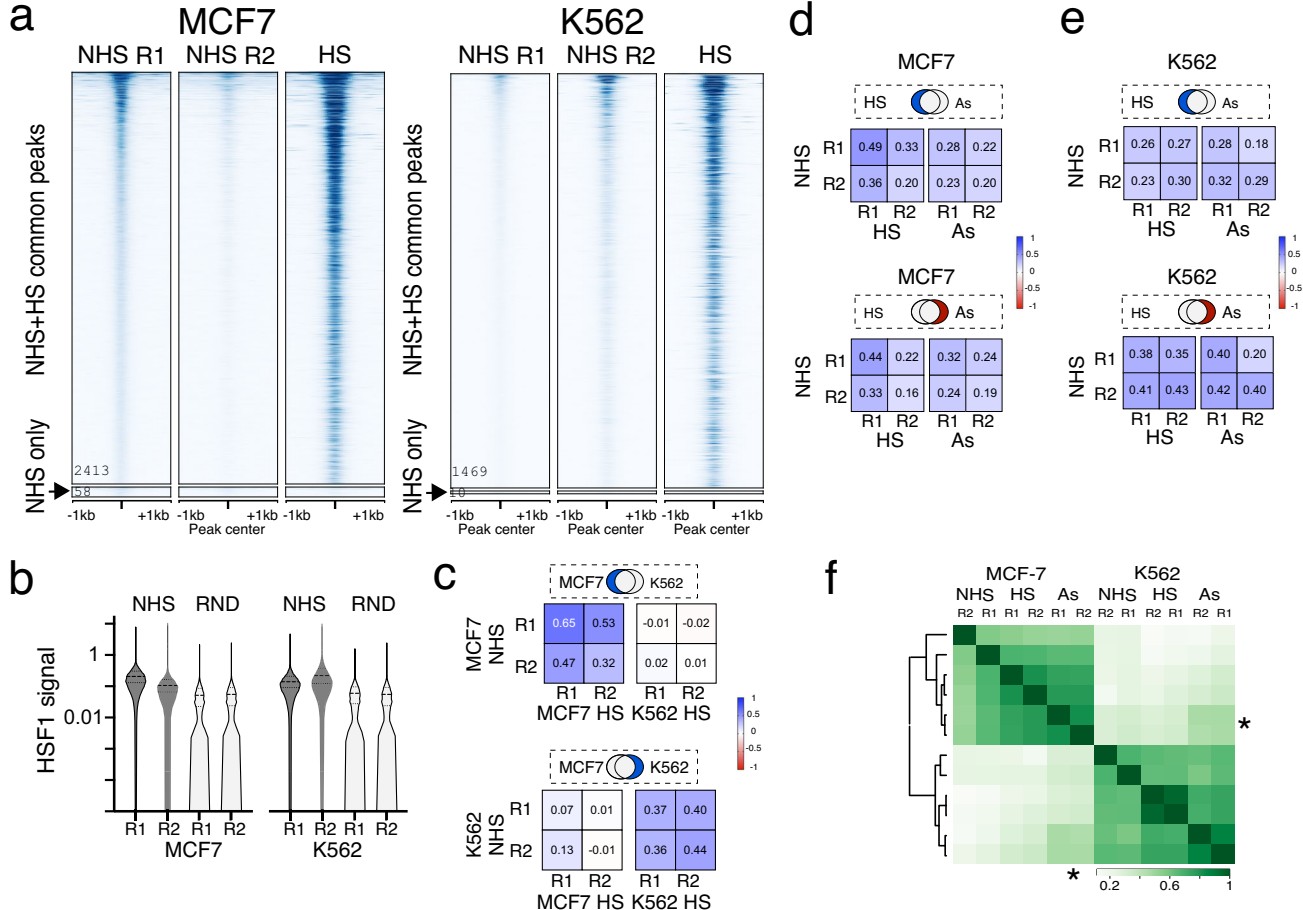

**Fig. 6 | HS-activated and basal HSF1 binding. a** Heatmaps of HSF1 peaks identified in NHS cells including peaks found in common with HS and a small number of peaks found exclusively in NHS cells (indicated by arrows), shown for two replicates. Heatmap for each cell line is sorted by NHS replicate 2 (R2) signal. A HS sample is included for comparison. **b** Averaged signal for NHS datasets from the locations of HS-only peaks not found in NHS samples (15,698 for MCF7 and 30,360 for K562 cells) compared to the signal for the same datasets from 29,036 randomly selected genomic locations of the same average size. Random locations were defined with bedtools random using the average size of HSF1 peaks and 30,000 locations were filtered against promoters, dTREs, and HSF1 peaks. HSF1 signal for HS-only peaks at NHS conditions is significantly higher than at random regions for each replicate

(Mann–Whitney test, two-tailed $p < 0.0001$). Boxplots show medians with top and bottom quartiles, and whiskers indicate 10–90th percentiles. **c** Comparison of basal and activated HSF1 signal for peaks found only in one cell line, as indicated in the mock Venn diagram above. Spearman correlation was calculated for cell line-exclusive HSF1 peaks between NHS and HS samples for the same and the opposite cell line. **d** Comparison of HSF1 signal for HS or As treatment-exclusive HSF1 peaks for MCF7 cells. **e** Same as (**d**), but for K562 cells. **f** Unbiased clustering of all HSF1 peaks across all datasets for each replicate. Individual samples are labeled post-factum. Asterisks mark the quadrant with correlation between MCF7 As and K562 As samples, which is higher than for any other conditions between cell lines. Source data are provided as a Source Data file.

NHS samples separated by the cell line (Fig. 6f), indicating that despite the stimulus-specific differences, HSF1 signal amplification in HSR is overall uniform. That the NHS samples showed cell line separation despite the low signal indicates that HSF1 binding patterns retain their identity across a broad dynamic range of stimulus intensity.

## HSF1 further opens remote inactive sites

Because chromatin defines how transcription factors interact with the genome[34,35], we related HSF1 signal to chromatin accessibility by examining ATAC and HSF1 data. A positive control *HSPH1* gene showed an HSR-dependent increase in ATAC signal along the gene body consistent with highly active transcription (Figs. 7a, 1a). However, HSR induced no drastic ATAC changes globally, as signal at the locations of peaks uniquely called in any one dataset was visually apparent throughout (Fig. 7b). ATAC peaks statistically different in HSR compared to NHS cells ($p$adj < 0.05) mostly increased in HS and decreased in As, with HSF1 associated with an increase in ATAC signal in either treatment (Supplementary Fig. 7a). A higher proportion of upregulated ATAC peaks was associated with HSF1 in HS than in As (Fig. 7c). ATAC signal in NHS cells was higher at promoters ($p < 0.0001$), but showed more prominent changes at intergenic regions (Fig. 7d). An

increase in ATAC signal at promoters of activated genes was mirrored by the increase in Pol II, but not H3K4me3 signal, and was not associated with HSF1 binding (Supplementary Fig. 7b, c). We conclude that HSF1 contributes to further chromatin opening at intergenic regions.

To identify the preference of HSF1 for pre-existing chromatin, we cross-examined HSF1 and ATAC signal. HSF1 and ATAC peaks overlapped at known functional elements including promoters and intergenic elements (dTREs and enhancers) (Supplementary Fig. 7d, e) so that most HSF1 peaks were found outside of these regions (Supplementary Fig. 7e, f). HSF1 thus appeared to bind to closed chromatin without apparent ATAC signal enrichment (Fig. 7e, f). However, careful examination revealed low-level ATAC signal in NHS cells around the locations of HSF1 peaks that would be acquired in HS (Fig. 7f, inset), consistent with HSF1 favoring pre-existing accessible sites. However, the magnitude of pre-existing ATAC signal did not explain HSF1 binding. For example, ATAC signal was higher at promoters, but HSF1 signal was higher at dTREs ($p < 0.0001$) (Fig. 7g). Likewise, HSF1 signal outside of functional elements appeared disproportionally higher at more closed sites compared to ATAC signal at the same sites (Fig. 7g, h). Accordingly, we noted a higher prevalence of HRE motifs among HSF1 peaks at more closed sites (Supplementary Fig. 7g). The

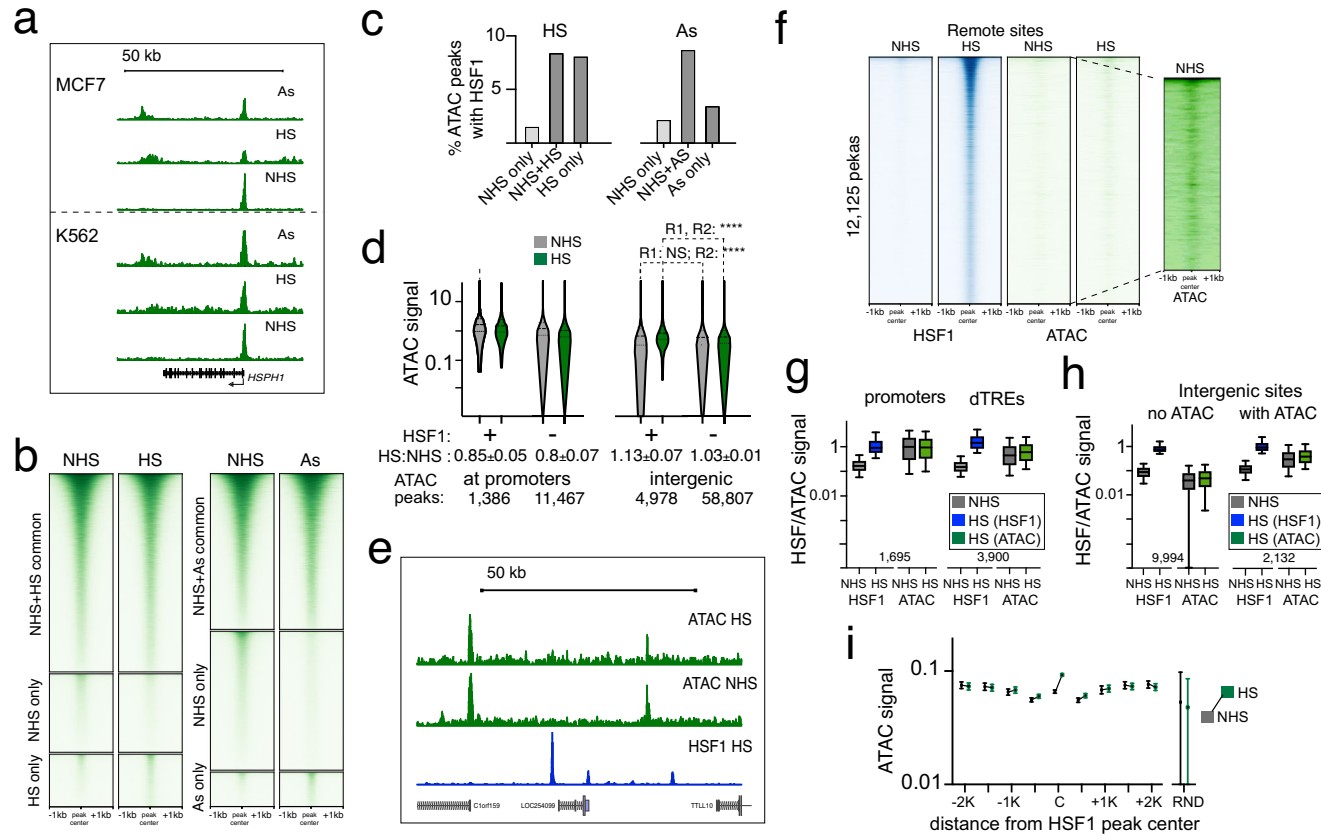

**Fig. 7 | HSF1 favors open chromatin. a** UCSC browser ATAC tracks of *HSPH1* gene region for indicated treatments. **b** Heatmaps showing common and treatment-unique ATAC peaks in MCF7 cells between NHS and indicated treatments. **c** Fractions of ATAC peaks overlapping MCF7 HSF1 peaks among ATAC peaks for common and treatment-unique categories defined in (**b**). **d** ATAC signal for the indicated groups of peaks based on the overlap with promoters and HSF1 peaks. Numbers below the graph indicate the number of peaks in each category and the ratios of HS: NHS signal based on two independent biological replicates (mean and range). Mann–Whitney test is shown for intergenic ATAC peaks (+/− HSF1) for each replicate. Significance (****) indicates $p < 0.0001$, and NS – $p = 0.1394$. **e** UCSC browser shot of a randomly chosen ~80 kb region showing ATAC and HSF1 signal in MCF7 cells. **f** Heatmaps showing HSF1 and ATAC signal at remote HSF1 peak sites outside of promoters, dTREs and annotated enhancers, for MCF7 cells. The inset

shows a contrast-adjusted signal for NHS ATAC panel indicating low-level pre-existing ATAC signal. **g** HSF1 and ATAC signal for promoters and dTREs for NHS and HS datapoints. **h** HSF1 and ATAC signal for remote genomic regions as in (**f**) that do (with ATAC) or do not (no ATAC) overlap with ATAC peaks. Signal in (**g**) and (**h**) is shown in the same scale. The numbers of individual loci used for each analyses are shown within the graph. The box and whisker plots show the medians with 10–90 percentiles. Numerical similarity between HSF1 and ATAC signal is coincidental. **i** ATAC signal at the 9732 distant MCF7 HSF1 peaks acquired in HS, shown as the mean with error bars defining the 95% confidence interval. ATAC signal centered on HSF1 peaks is shown for 500-nt bins around the HSF1 peak center ("C") with the offset distance from the center indicated, up to +/− 2 K (kb). ATAC signal from random regions is also shown. NHS ATAC signal is indicated in black and HS in green. Source data are provided as a Source Data file.

binding of HSF1 to less accessible chromatin may be more reliant on direct DNA sequence-specific interactions.

Lastly, we examined remote HSF1 peaks for changes in chromatin accessibility induced in HS. These sites showed an ATAC signal increase in HS at the exact locations of HSF1 peaks, but not in the immediate surrounding regions (Fig. 7i). This increase in ATAC signal was modest as it did not result in ATAC peaks (Fig. 7f). Nevertheless, co-occurrence of this increase with HSF1 binding locations lends support to HSF1 binding independent of the ChIP method. These data also suggest that HSR can trigger changes in remote genomic regions.

## Discussion

Challenging distant cell lines with different stimuli that activate a common transcription factor allowed us to characterize its genome-wide binding and connection to nearby transcription. Earlier work noted distinct HSF1 patterns between cancer cells and noncancer cells responding to heat shock[4]. However, despite the growing numbers of available TF binding datasets[25], information on genome-wide binding of transcription factors between cell types, conditions and basal versus activated states in any system remains limited. Using HSR as a model, we show that cell type specific HSF1 patterns robustly retain their identity between basal conditions and various degrees of activation. In contrast to *Drosophila*, in which transcription in heat shock is globally inhibited in favor of few highly activated loci[1,15,16], mammalian transcription continues across the genome. The numbers of peaks and upregulated genes in our data are consistent with observations for other TFs[25] and are in line with other environmental and physiological responses[36–39]. Whether induced by heat or chemically, mammalian HSR is not quantitatively different from responses to other signals. TF binding patterns reflecting the activity of signaling pathways[37–39] may serve as readouts of cellular identity.

HSF1 binding at *HSP* gene promoters prior to activation was previously noted on individual genes[17,40] and in global datasets[25]. We find that despite the differences in signal intensity, genome-wide HSF1 patterns show similarity before and during HSR. It is conceivable that low-level basal HSF1 binding (Fig. 6) may have resulted from inadvertently activating HSR during routine cell culturing. However, our basal readouts are in line with those reported previously[4,8,9]. Low-intensity HSF1 binding indicates that HSR is not an all-or-nothing response, but instead is commensurate with the magnitude of a stimulus. Retention of genome-wide patterns under different magnitudes of HSR suggests that HSF1 is distributed across the genome proportionally under a broad range of active HSF1 concentrations in the cell. This is consistent with rapid sampling of available sites by individual TF molecules[41,42]. Previous work suggested that HS causes no major changes in the nuclear architecture[31,43]. Our ATAC data show further that the architecture is preserved in HSR down to individual loci. Even though how exactly HSF1 binding patterns echo the nucleus remains to be defined, the binding of HSF1 to existing functional elements such as promoters or enhancers appears to be driven to a higher extent by co-factors or favorable DNA structure whereas the binding to remote sites outside of functional elements relies more on the DNA sequence.

Comparing HSR across cell lines and stimuli shows that known HS-responsive *HSP* genes are activated as a conserved HSF1-dependent cohort[8,22]. Outside of these genes, however, HSF1 binding is largely decoupled from transcription activation. This decoupling is due to variability of transcriptional outcomes rather than HSF1 binding, and likely reflects pervasive regulation of transcription by combinations of transcription factors. Apart from HSF1, response to heat has been shown to involve transcription factors including HSF2, SRF, CTCF, ER-α, among others[7–9,44–46], and both HS and As may involve oxidative stress components[6,47,48]. Sequence search around HSF1 binding sites points to enrichment of AP-1 FOS-related components among

HS-specific HSF1 peaks (Supplementary Fig. 2c), indicating possible involvement of AP-1 in shaping HSF1 binding under some stimuli. Variable gene activation under common HSF1 binding (Fig. 3) is consistent with individual TFs providing a stable context for combinatory regulation. This notion is corroborated by recent yeast work showing relatively low incidence of mutual co-dependency of TF binding[49]. In human cells, deletion of HSF2 does not appear to alter HSF1 binding[9]. However, despite the overall cell type specificity, a portion of HSF1 binding is dynamic. First, HSF1 can bind to treatment-specific sites in the same cell line (Fig. 3), showing detectable correlation in As treatments between distant cell lines (Fig. 6f). These findings indicate that heat-independent HSF1 binding, and possibly function, while limited in scope, are conserved. Second, a recent study reported changes in TF binding patterns over time[50]. While the underlying mechanisms remain to be determined, these changes may represent early stages of cell state transitions occurring in response to signals.

The binding of HSF1 at remote genomic sites suggests potential function outside of transcription. At least some of low-affinity TF binding sites are likely functional[51], with cooperativity among individual binding events driving the binding across the genome including transcriptional enhancers[52,53]. HSF1 has been previously implicated in DNA repair via the PARP complex[54]. TF binding may also nucleate the opening of closed genomic regions, whether immediately, over time or with repeated stimulation. HSR is often ectopically activated in cancers. Arsenic is a transforming agent that in addition to mutations can induce epigenetic changes[55], and heat has been recently shown to induce transcriptional memory[56]. During this manuscript revision, a new study reported distinct consequences of Wnt/β-catenin activation between a cancer cell line and embryonic stem cells[50], highlighting a pivotal role of the cellular context in defining chromatin plasticity. Changes prompted by stimuli should depend on transcriptional responses to activate transcription factor binding to DNA, but not necessarily transcription[57].

## Methods

### Cell culture and treatments

All cells were purchased from the American Type Culture Collection and used within the first 15 passages. Cells were grown in 15 cm dishes as before[58] and growth media was replaced with fresh media 24 h before treatments[31,59]. Heat-Shock treatment was started by placing a dish onto a heated water surface for 1 min to raise the temperature quickly and continued in a 42 °C 5% $CO_2$ incubator for the remaining time. Arsenic treatment was performed by the addition of 500 μM Sodium meta-arsenite (Sigma) and incubation at the ambient 37 °C under 5% $CO_2$[26].

### Western blotting

Cells were scraped (MCF-7) or collected from suspension (K562), washed with ice-cold Phosphate-Buffered Saline (PBS), resuspended in lysis buffer (8 M Urea, 1% SDS, 50 mM Tris pH 6.8), and protein concentration was measured on Qubit fluorometer using protein assay (Invitrogen). Approximately 30 μg of total protein was resolved on a 10% SDS-PAGE gel and transferred onto a PVDF membrane. Non-specific binding was blocked with 5% nonfat milk diluted in 1x Tris-Buffered Saline pH 7.5 with 0.1% Tween-20 (TBS-T) followed by primary antibody incubation overnight and three TBS-T washes. Blots were developed using horseradish peroxidase-conjugated secondary antibody (GE #NA934) and imaged on a Li-COR Odyssey Fc imager. Western blotting was performed against HSF1 (Enzo #ADI-SPA-901-D) or [pSer$^{326}$]-HSF1 (Enzo #ADI-SPA-902-D) antibodies and GAPDH (Millipore #ABS16) as control. All antibodies used for western blotting were at 1:1000 final dilution. Band sizes were verified using Precision Plus dual color protein ladder (Bio-Rad). For an example of presentation of full scan blots, see the Source Data file.

## Reverse transcription and quantitative PCR

Approximately 500 ng of total RNA was used to synthesize cDNA using random hexamers and SSRT III reverse transcriptase (Life Technologies). PCR primers were from IDTDNA and were synthesized in 25 nmol scale with no additional purification (Supplementary Data 8). Quantitative PCR data were normalized against GAPDH gene transcripts and shown as fold change from at least three independent biological replicates.

## ChIP-sequencing

Approximately $2 \times 10^7$ cells were crosslinked with 1% ethanol-free formaldehyde (Thermo) in serum-free DMEM/F12 media at 25 °C for 10 min, quenched with 125 mM Glycine, and resuspended in 1 ml of Lysis buffer (1% Sodium-Dodecyl Sulfate (SDS), 50 mM Tris (pH 8.0), 10 mM EDTA, 1X PMSF and protease inhibitor cocktail) followed by disruption in the Covaris S2 sonicator for 6 min (with peak power = 140, Duty factor = 5, Cycles/burst = 200). ChIP-sequencing was performed with anti RPB1 NTD (D8L4Y) (CST #14958), HSF1 (Enzo #ADI-SPA-901-D) or H3K4me3 (Abcam #ab8580) antibodies. HSF1 antibody (SCBT #sc-9144, no longer available) was used for some controls as indicated. Each chromatin sample for ChIP-sequencing contained 150 μg's worth of DNA (usually 100–250 μl) as extrapolated by extracting DNA from 1% of the sample. The sample volume was adjusted to 2 ml with Dilution buffer (1% Triton X-100, 2 mM EDTA, 20 mM Tris pH = 8.0, 150 mM NaCl, 1 mM Phenylmethylsulfonyl fluoride (PMSF) and cOmplete protease inhibitor cocktail (Sigma), 1 tablet per 50 ml). Protein A and Protein G Dynabeads (Thermo) premixed at 1:1 ratio (50 μl) were added for pre-clearing for 1 h at 4 °C. After removing the beads on a magnetic rack, one percent of each pre-cleared sample was taken as input. To the rest of the reaction, 5 μg of an antibody was added (adjusted based on antibody concentration, typically 1:400 dilution). After overnight incubation at 4 °C under slow rotation, 30 μl of Protein A + G bead slurry was added and reactions were incubated for an additional 2 h, followed by washing with high salt (20 mM Tris-HCl pH 8.0, 2 mM EDTA, 500 mM NaCl, 1% Triton X-100, 0.1% SDS), LiCl (Tris-HCl pH 8.0, 2 mM EDTA, 250 mM LiCl, 1% Igepal and 1% sodium deoxycholate), low salt (Tris-HCl pH 8.0, 2 mM EDTA, 150 mM NaCl, 1% Triton X-100, 0.1% SDS) wash buffers, Tris-EDTA (TE) buffer pH 8.0, on a magnetic rack, and elution with two changes of Elution Buffer (1% SDS, 10 mM EDTA, 50 mM Tris-HCl pH 8.0), 200 μl each, followed by ethanol-precipitation[58,60]. Four percent of DNA after ChIP was taken for validation by qPCR using positive and negative control primers (Supplementary Data 8) to calculate the percent input. DNA libraries were prepared from the ChIP sample using NEBNext Ultra II DNA library kit (New England Biolabs) protocol without size selection. Libraries were prepared by PCR amplification of cDNA using TruSeq Small RNA PCR primers (Illumina), 250 μM dNTP mix, 1x HF Phusion buffer, 1 M Betaine, and Phusion DNA polymerase (NEB). Reactions were supplemented with 1xEvaGreen dye (Biotium) to monitor amplification and were manually stopped within the linear range of amplification, normally reached after 12 to 15 PCR cycles. Libraries were additionally validated by qPCR with the same primers (Supplementary Data 8)[60].

## PRO-sequencing

Precision Run-On sequencing (PRO-seq)[61] was done with ~$2 \times 10^7$ cells collected by scraping (for MCF7 cells) or centrifugation (for K562 cells), resuspending in 20 ml Permeabilization buffer (10 mM Tris-HCl, pH 7.5, 300 mM sucrose, 10 mM KCl, 5 mM MgCl$_2$, 1 mM EGTA, 0.05% Tween-20, 0.1% NP40 substitute, 0.5 mM DTT, protease inhibitors cocktail (Sigma) and RNaseIn RNAse inhibitor (Thermo)), washing in the same buffer and resuspending in the final volume 100 μl of Storage buffer (10 mM Tris-HCl, pH 8.0, 25% (v/v) glycerol, 5 mM MgCl$_2$, 0.1 mM EDTA, 5 mM DTT), flash-freezing in liquid N$_2$ and storing at −80 °C until use. Run-on reactions were carried out in a 37 °C water bath by adding an equal volume (100 μl) of run-on mix (10 mM Tris-HCl, pH 8.0, 5 mM MgCl$_2$, 300 mM KCl, 1 mM DTT, 11-biotin-labelled ribonucleotides [Perkin Elmer, 3 μl each, undiluted for ATP and GTP, and diluted 1:5 for CTP and UTP], RNAseIn RNAse inhibitor and 0.5% sarkosyl) for 3 min. Reactions were immediately mixed by pipetting with a cut-off 200 μl pipette tip. Reactions were stopped with 300 μl Trizol-LS followed by extraction of RNA. Water phase was additionally extracted with 0.6 volumes of chloroform to remove traces of phenol before ethanol-precipitation. RNA was fragmented by incubating in 0.2 M NaOH on ice for 10 min, neutralized with an equal volume of 1 M Tris-HCl pH 7.0, purified on a Bio-Rad P-30 Micro BioSpin desalting column, and biotin-enriched using 30 μl of C1 Streptavidin magnetic beads (Thermo) in the final volume of 100 μl of Binding Buffer (10 mM Tris-HCl, pH 7.5, 300 mM NaCl and 0.1% (v/v) Triton X-100), for 20 min at room temperature with gentle shaking. Beads were washed with Binding, High-Salt, Low Salt buffers and TE, and RNA was recovered by double extraction with regular Trizol reagent (300 μl each time), using an additional chloroform extraction of the combined aqueous phase to remove traces of phenol prior to adding GlycoBlue (1 μl, Ambion) and ethanol precipitation with 3 volumes of 96% ethanol and incubation for at least 10 min at room temperature. After microcentrifugation at max speed for 15 min at 4 °C and 70% ethanol wash, air-dried ethanol-free RNA pellet was resuspended in 4 μl of 2.5 mM VRA3 3'-small RNA adapter and ligated by adding a mixture containing 1 μl T4 RNA ligase buffer, 1 μl 10 mM ATP, 2 μl 50% PEG 8000, 1 μl T4 RNA ligase 1 (all NEB) and 1 μl RNAseIn RNAse inhibitor, followed by ligation at 20 °C for 6 h. Following ligation, the reaction was biotin-enriched as above, treated with T4 Polynucleotide kinase (PNK) (NEB) containing 1 mM ATP in 10 μl of 1x PNK buffer (NEB) for 30 min at 37 °C followed by addition of 90 μl of RNA decapping mix (containing 20 μl of 5x Thermopol buffer and 1 μl of RppH (both NEB) and 1 μl RNAseIn RNAse inhibitor) and incubation for an additional 60 min. Following Trizol extraction, RNA was resuspended in 4 μl of 2.5 mM VRA5 5'-adapter and ligated as above. After another biotin enrichment, RNA was reverse-transcribed using SuperScript III (Thermo) reverse transcriptase with RP1 primer, and reactions were amplified using real-time PCR in the presence of EvaGreen dye as above. PRO-seq oligonucleotides (Supplementary data 8) were from IDTDNA synthesized in 100 nmol scale and HPLC purification. Amplified libraries were run on a 6% TBE gel (Novex) in 1X Tris-Borate-EDTA (TBE) buffer. The gel was stained with ethidium bromide, visualized with a 312 nm UV transilluminator, and areas between ~125–300 bp DNA sizes were cut out and extracted using crush and soak method[62]. Specifically, gel slices were transferred into a gel-breaker tube (IST Engineering) placed inside a capless 2 ml microcentrifuge tube and spun at max speed to crush and extrude gel material. Elution was done by soaking the gel slurry in 400 μl of TE containing 300 μM NaCl for 2 h at room temperature with gentle shaking. The slurry was transferred into a 0.22 μm spin Filter for Gel Matrix (Agilent) and microcentrifugated at 1800 g for 2 min. DNA from the gel-depleted eluate containing the library was ethanol-precipitated and resuspended in 20 μl of TE, quantified and Illumina-sequenced (Psomagen or Yale Center for Genome Analysis).

## ATAC-sequencing

The Assay for Transposase Accessible Chromatin with high-throughput sequencing (ATAC-seq) was performed using 50,000 cells and the original Tn5 buffer (10 mM Tris-Cl, pH 7.4, 10 mM NaCl, 3 mM MgCl$_2$ and 0.1% Igepal)[63]. Incubation with Tn5 transposase enzyme (Diagenode, 2.5 μl per 25 μl reaction) was done in a shaking heating block at 37 °C and 500 RPM for 30 min followed by real-time PCR amplification with Nextera Illumina-compatible dual index primers and NEBNext High-Fidelity 2x PCR mix with Evagreen dye to avoid overamplification.

## ChIP-seq data analysis

Sequencing was done to the depth indicated (Supplementary Data 1). Illumina adapters were removed from raw files with trimmomatic using paired end mode and keepBothReads set as 'true' and aligned to hg19 genome with hisat2 using --no-spliced-alignment and otherwise default parameters[64]. HSF1 bam files were scaled to the lowest coverage sample using samtools. BigWig files were made using bamCoverage[65]. For all ChIP-seq data except our HSF1 ChIP-seq data, read counts were normalized to the sequencing depth and signal was calculated from the bigwig files using multiBigwigSummary with sequencing depth coverage normalization. For HSF1 peak calling, alignments were pre-filtered against blacklisted genomic regions that included ENCODE ENCSR636HFF excluded list regions[66], regions known to be absent (zero-copy number) in either MCF7 or K562 cell lines from Cancer Cell Line Encyclopedia[67], and UCSC blacklisted human genome regions[68]. After removing PCR duplicates with Picard's MarkDuplicates, and peaks were called using macs2[69]. UCSC genome browser tracks were generated using sequencing depth-normalized bigwig files, with each type of a track shown in the same numerical y-axis scale for all samples. Individual peak list replicates were merged by combining peaks present in both replicates (K562 cells) or either replicate (MCF7 cells). Differential signal was calculated using DiffBind[70]. PCA plots were generated using affinity values default function for PCA in DiffBind. The 23 K ($n = 23,698$) gene list and exact genes and TSS definitions were as defined previously[31], with promoter regions defined as TSS + / −1000 nt. Venn diagrams were drawn using R venneuler package. Motif search for HSF1 was done with findmotifs.pl (HOMER)[23].

## PRO-seq and ATAC-seq data analysis

Illumina small RNA TruSeq adapters were trimmed and reads shorter than 15 nt were discarded. Trimmed reads were used to remove ribosomal RNAs and remaining reads were aligned to hg19 using hisat2 using no-splicing option. Individual samples were normalized using 3'-ends of long genes[61]. Differentially expressed genes were defined using DESeq2. Identification of dTREs was done using the remote site[31] with default parameters for individual replicates, retaining dTREs present in both biological replicates for MCF7 cells and one replicate for K562 cells. HS-specific dTREs were defined as present only in HS and not in NHS cells. For ATAC-seq, adapter-trimmed reads were aligned to hg19 genome as above, followed by converting to BigWig format using bamCoverage with the default bin size of 50 bp. Duplicate reads were removed using MarkDuplicates and the signal was normalized by counts per million (CPM). Heatmaps were generated using plotHeatmap from deepTools.

## Statistics and reproducibility

Experiments are based on a minimum of two repeats with similar results, with quantitative PCR using three or more independent biological replicates. Differentially expressed genes were defined using DESeq2 based on $p$_adj < 0.05. ChIP-seq peaks were defined using macs2 based on $q$ value cutoff of 0.01. Omics datasets were visualized as violin plots or box plots indicating top and bottom quartiles, and error bars showing 10−90 percentiles. Comparisons were done using nonparametric Mann−Whitney test calculating two-tailed $p$-values.

## Reporting summary

Further information on research design is available in the Nature Portfolio Reporting Summary linked to this article.

## Data availability

The sequencing data generated in this study have been deposited in the Gene Expression Omnibus (GEO) database under accession code GSE209687. All data used in this study are based on hg19 human genome assembly. Public ChIP-seq datasets analyzed during the current study are under GEO accession codes GSE85158, GSE105028, GSE38912, and GSE43579 [https://www.ncbi.nlm.nih.gov/geo/query/acc.cgi?acc=GSM2367735]. Source data are provided with this paper.

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

## Acknowledgements

This work was supported by the National Science Foundation CAREER award 1750379 to N.S. and by the National Institute of General Medical Sciences of the National Institutes of Health under Award Numbers U54GM128729 and 2P20GM104360. The funding bodies had no role in the design of the study, collection, analysis, and interpretation of data or in writing the manuscript. We thank Archana Dhasarathy, Motoki Takaku, Min Wu, Benjamin Roche and Paul Wade for critical reading of the manuscript, members of Dhasarathy and Takaku labs for vigorous discussions, as well as Sara Apostal and Mike Hill for handling omics libraries and data. We are grateful to the UND Genomics Core and Yale Center for Genome Analysis for prompt service.

## Author contributions

S.G.D. designed the study, performed experiments to generate most datasets, analyzed and interpreted data, and wrote the manuscript. B.D.K. analyzed HSF1 ChIP-seq and ATAC data for differential signal and interpreted the analyses. B.L. performed Western blots. D.Pa., D.Pe., M.V. and A.T. analyzed PRO-seq data. N.K.K.K. prepared cells for PRO-seq experiments. A.A. performed HSF1 ChIP-seq experiments with SCBT antibody. N.S. designed the study, analyzed data and wrote the manuscript.

## Competing interests

The authors declare no competing interests.
