## [Peer Review File · Nature Communications]

Transcriptional Responses of Cancer Cells to Heat Shock-Inducing Stimuli Involve Amplification of Robust HSF1 Binding.REVIEWER COMMENTS

Reviewer #1 (Remarks to the Author):

In this manuscript, Dastidar et al. asked what extent the genome-wide binding of HSF1 during HSR varies by the cell type versus the stimulus, and how this binding relates to nascent transcription. They performed ChIP-seq for HSF1, Pol II, and H3K4me3 (transcription factor binding and a histone mark), PRO-seq (nascent transcription), and ATAC-seq (chromatin accessibility), using human cancer MCF-7 and K562 cells untreated (NHS) and treated with heat shock (HS) and arsenite (As). The authors identified large numbers of HSF1 ChIP-seq peaks in these cells, and analyzed these data with RPO-seq and ATAC-seq data.

They showed widespread HSF1 binding (promoters and distal regions) in HS- and As-treated MCF-7 cells, and suggested that a part of these HSF1 binding was correlated with gene activation and a majority of activated genes were associated with common HSF1 binding (Figures 1-3). The authors then compared HSF1 binding and gene activation in two cell lines MCF-7 and K562 cells, and claimed cell-type specificity in HSF1 binding locations and intensity (Figure 4). Connection between HSF1 binding in the intergenic regions and gene activation modestly existed in both cell lines, like connection between HSF1 binding in the promoters and gene activation (Figure 5). The authors further claimed that HSF1 binding signals under NHS condition were amplified during HS condition in MCF-7 and K562 cells, although HSF1 signals in NHS MCF-7 and NHS K562 differed (Figure 6). Lastly, the authors showed that ATAC signals at promoters were high in NHS cells and were not changed so much during HS, suggesting that HSF1 favors open chromatin (Figure 7). On the other hand, HSF1 binding during HS contributed chromatin opening at intergenic regions (Figure 7).

The authors conducted a lot of genome-wide analyses with high quality. Large amounts of data are adequately presented and support most conclusions about connection between HSF1 binding and gene activation before and after treatment of HS or As in two cancer cell lines. This work makes an important contribution to the fields of the heat shock response and transcriptional regulation. Some minor points should be clarified before publication.

Comments:

1) The title; "Transcriptional responses of cancer cells to heat shock-inducing stimuli involve amplification of robust HSF1 binding." It seems to be one (in the text at P12, Fig. 6) of several conclusions in this manuscript, and does not match well with the statement in the abstract. In the abstract, the authors do not mention "cancer cells" at all, and do not state major implication of this study (probably related to the title). It would be better to reconsider the title and/or abstract.

2) In the discussion section, the authors did not discuss about the topics that "Transcriptional responses of cancer cells to heat shock-inducing stimuli involve amplification of robust HSF1 binding" What was known about HSF1 binding sites previously in cancer cells? Were HS-induced HSF1 binding sites determined before stimuli? Did HSF1 binding in these sites not detected in previous studies? Why were

they able to find the results leading to this conclusion in this study (technical aspect)? Please discuss these points understandable to non-specialist readers.

3) The authors stated “~18,000 peaks ($q < 0.01$) identified between two independent biological replicates (Supplementary Fig. 2a; Supplementary File 2)” (P5). The number of identified HSF1 peaks in HS MCF-7 cells is high compared with those in previous reports. The authors should show Venn diagram for HSF1 peaks between two experiments (in both NHS and HS MCF-7 cells) in Supplementary Figure. Please indicate the number of identified HSF1 peaks in NHS and HS MCF-7 cells in the text. Is HSF1 binding intensity from one experiment used for generation of heatmap in Figure 1c?

4) The authors stated “An active promoter histone mark H3K4me3 showed no drastic changes in HS compared to untreated control (non-heat shock, NHS) cells (Fig. 1a, d)” (P4). Is this data consistent with previous reports (H3K4me3 level in a specific HSP gene and genome-wide H3K4me3 analysis)?

5) The authors stated “Elevated temperature is therefore not a prerequisite for genome-wide HSF1 binding” (P6). The heading of Fig. 2 “Fig. 2: Widespread HSF1 binding is independent of elevated temperature.” (P37). Is it better to state that widespread HSF1 binding is induced by As treatment?

6) What “As-exclusive” and “As-specific” mean (P7, P8)?

7) In line 168 (P6), “within a cell line (Fig. 4b; Fig. 2f).” Figure 4c should be cited here.

8) Figure 2a and 2b; amend color of boxes.

9) Figure 4b; gene names and structures are too small to see.

10) Figure S6; it is better to add heading of this figure.

Reviewer #2 (Remarks to the Author):

This study profiles HSF1 binding and gene expression under heat shock response in two conditions and two cell lines. The authors found that HSR activation is marked by widespread elevation of HSF1 binding. HSF1 binding locations are shared between HSR induced through temperature and arsenic stimuli, but activated genes are stimulus-specific. HSF1 binding is enriched at activated genes, but most HSF1 binding does not activate expression. HSR induced HSF1 binding and gene activation are distinct between MCF7 and K562.

Major limitations:

1. The motivation of this study was not justified in the Introduction section. As molecular profiling of heat shock response has been performed in multiple cell lines and conditions (for example, Mahat et al. 2016, Vihervaara et al. 2017), the significance of this study remains unclear. Results in Fig1 (including

elevated HSF1 binding in HSR) and Fig 3 (decoupling between HSF1 binding and gene activation) can be found from multiple similar studies.

2. Mechanistic and functional analysis are largely missing from the current manuscript. Several interesting questions arise from the data but the authors didn't attempt to address them. For example, why do arsenic stimuli activate a distinct set of genes from high temperature? Why do increased HSF1 binding activate some genes but not the other? Gene ontology can be applied to uncover gene programs underlying treatments and cell type specific HSR. Motif enrichment analysis can be performed to reveal co-factors of HSF1 to better understand regulatory mechanisms. Unfortunately, such in-depth analysis can rarely be found, as current analysis focuses on describing the data rather than interpreting it. New hypotheses were not raised from the data or tested either.

Reviewer #3 (Remarks to the Author):

In the current manuscript, Dastidar and colleagues attempted to understand the relationship between HSF1 binding genome-wide and the consequence of gene expression in two different carcinoma cell lines in response to two different external stimuli, temperature elevation, and toxic arsenite supplementation. To investigate the relationship, the authors performed multi-omics, including ChIP-seq, PRO-seq, and ATAC-seq.

The authors designed and executed their planned experiments systematically, and the manuscript is written well. However, this study does not add ample information to the existing finding except for a head-to-head comparison of genome-wide HSF1 binding and differential gene expression between two cell types in response to two different stimuli. Moreover, this study lacks identifying potential molecular mechanisms for cell type- and the stimuli-specific connection between HSF1 binding to DNA and differential gene expression. Therefore, this reviewer thinks this manuscript, in its current content and format, has limited scope in advancing the respective field.

However, this reviewer recommends addressing some concerns to improve the quality of the manuscript in a future submission.

1. Determine and compare side-by-side HSF1 occupancy at NHS and HS conditions for the promoters of genes upregulated explicitly upon heat shock or arsenic treatment with the genes, the expression of which is unchanged.

2. Upon heat shock in MCF7 cells by PRO-seq, the authors identified the upregulation of 2,228 genes. However, I did not see anywhere in the manuscript that the authors analyzed how many HS and non-HS genes. Similarly, the same pieces of information are missing for As-induction. Moreover, the authors did not mention how many genes are upregulated upon As-treatment and how many genes are downregulated upon heat shock and As-treatment.

3. Line 114-117, Fig 2h, and Supplementary Fig 8: It is unclear whether the authors compare all activated genes or HSF1-bound activated genes between two conditions (HS and As-treatment). If these figures represent the comparison between all activated genes, then the authors should compare HSF1-occupied activated genes between the two treatments.

4. Line 117-119: In comment 3, if authors compared only HSF-bound activated genes, then the most relevant conclusion would be "altogether the results suggest depending on the type of upstream signals HSF1 activates different sets of genes in same cell type".

5. Line 126-128 and Fig 3b,c: The authors observed only 1/4th and 1/5th of activated genes' promoters are HSF1-bound in HS and As-treatment, respectively. The authors also identified approximately 1/3rd, and 1/4th of not activated genes' promoters are HSF1-occupied. How about the correlation between HSF1-unbound activated genes and HSF1-bound not activated genes in HS and As-treatments?

6. To understand the relation between HSF1 binding and chromatin accessibility, authors compared ATAC-seq data with HSF1-peaks in NHS and HS conditions. This reviewer wonders whether the chromatin accessibility data correlates well with the gene activation.

7. Sometimes the detailed description of the analysis challenges understanding and extracting the study's big picture.

8. The authors can consider combining the supplementary figures into fewer.

REVIEWER COMMENTS

We thank each reviewer for feedback that we believe helped us improve the manuscript.

Reviewer #1 (Remarks to the Author):

In this manuscript, Dastidar et al. asked what extent the genome-wide binding of HSF1 during HSR varies by the cell type versus the stimulus, and how this binding relates to nascent transcription. They performed ChIP-seq for HSF1, Pol II, and H3K4me3 (transcription factor binding and a histone mark), PRO-seq (nascent transcription), and ATAC-seq (chromatin accessibility), using human cancer MCF-7 and K562 cells untreated (NHS) and treated with heat shock (HS) and arsenite (As). The authors identified large numbers of HSF1 ChIP-seq peaks in these cells, and analyzed these data with RPO-seq and ATAC-seq data.

They showed widespread HSF1 binding (promoters and distal regions) in HS- and As-treated MCF-7 cells, and suggested that a part of these HSF1 binding was correlated with gene activation and a majority of activated genes were associated with common HSF1 binding (Figures 1-3). The authors then compared HSF1 binding and gene activation in two cell lines MCF-7 and K562 cells, and claimed cell-type specificity in HSF1 binding locations and intensity (Figure 4). Connection between HSF1 binding in the intergenic regions and gene activation modestly existed in both cell lines, like connection between HSF1 binding in the promoters and gene activation (Figure 5). The authors further claimed that HSF1 binding signals under NHS condition were amplified during HS condition in MCF-7 and K562 cells, although HSF1 signals in NHS MCF-7 and NHS K562 differed (Figure 6). Lastly, the authors showed that ATAC signals at promoters were high in NHS cells and were not changed so much during HS, suggesting that HSF1 favors open chromatin (Figure 7). On the other hand, HSF1 binding during HS contributed chromatin opening at intergenic regions (Figure 7).

The authors conducted a lot of genome-wide analyses with high quality. Large amounts of data are adequately presented and support most conclusions about connection between HSF1 binding and gene activation before and after treatment of HS or As in two cancer cell lines. This work makes an important contribution to the fields of the heat shock response and transcriptional regulation. Some minor points should be clarified before publication.

Comments:

1) The title; “Transcriptional responses of cancer cells to heat shock-inducing stimuli involve amplification of robust HSF1 binding.” It seems to be one (in the text at P12, Fig. 6) of several conclusions in this manuscript, and does not match well with the statement in the abstract. In the abstract, the authors do not mention “cancer cells” at all, and do not state major implication of this study (probably related to the title). It would be better to reconsider the title and/or abstract.

We intended HSF1 binding to be the take-home title point, with the abstract expanding on HSF1 binding in terms of potential function. We edited the abstract to address this query. Regarding “cancer cells” in the title, we had considered “human cells”, but decided against it to avoid implying primary cells or tumors. We now mention “cancer cells” in the abstract for consistency. To reflect the title point (robustness) more closely, we now emphasize that HSF1 binding patterns are retained between activated and basal states.

2) In the discussion section, the authors did not discuss about the topics that “Transcriptional responses of cancer cells to heat shock-inducing stimuli involve amplification of robust HSF1 binding” What was known about HSF1 binding sites previously in cancer cells? Were HS-induced HSF1 binding sites determined before stimuli? Did HSF1 binding in these sites not detected in previous studies? Why were they able to find the results leading to this conclusion in this study (technical aspect)? Please discuss these points understandable to non-specialist readers.

We agree with this point and now try to address it in the introduction and discussion sections. Early studies raised some key questions about global HSF1 patterns. One is potential differences in genome-wide binding between cell types (Mendillo et al, 2012). Another is relationship between HSF1 binding and transcription (Mahat et al., 2016), wherein disconnect between HSF1 binding and nascent transcription activation was noted. To the best of our knowledge, HSF1 patterns were not compared between different cell lines beyond noting higher signal in HS versus NHS in noncancerous cells (such as Mendillo et al., 2012, Figure 3, and Vihervaara 2013). Cancer cell lines were not assessed during HS in that study, but a later work (Vihervaara 2013) showed HS response in K562 cancer cells.

Figure R1. Comparison of NHS HSF1 peaks between studies. A. MCF7 cells based on datasets from current work (GD) and Mendillo et al 2012. B. Comparison of our MCF7 dataset with the closest available NHS dataset – MDA231 from Smith et al., 2022). The top five hits sorted by p-value from Homer Known Motifs output shown for each slice of the corresponding Venn diagram. HRE motifs are indicated on the left and percentage of peaks with HRE motifs in larger font.

In terms of datasets, for MCF7 cells, we are aware of one that contains NHS-only samples (Mendillo et al. 2012). Comparing MCF7 NHS data from us and Mendillo et al using our reanalyzes, we noted ~640 HSF1 peaks (Figure R1A). We find enrichment of HSF1 on promoters of all genes listed in figures for cancer and noncancer cell lines in Mendillo et al 2012 in HS, except LY6K (not shown). Between Mendillo 2012 NHS and our NHS datasets, we find enrichment of HSF1 HRE motif in overlapping and in our unique peaks (Figure R1A). We also know of recent NHS-only HSF1 data by Mendillo group for several cell lines, not MCF7 or K562 (Smith et al 2022). Looking at Smith et al MDA231 NHS data, we find HRE motifs in all groups of peaks overlapping with our MCF7 NHS data, with low overlap between datasets

consistent with cell type specificity of binding (Figure R1B). Regarding HSF1 binding before heat shock, low-level binding of HSF/HSF1 to HSP gene promoters is evident in all public NHS datasets, and is also noted in *Drosophila* (for example, Guertin and Lis, 2010 and Gonsalves et al, 2011). In response to this query, we modified the discussion and introduction sections to indicate the prior state of knowledge.

3) The authors stated “~18,000 peaks ($q < 0.01$) identified between two independent biological replicates (Supplementary Fig. 2a; Supplementary File 2)” (P5). The number of identified HSF1 peaks in HS MCF-7 cells is high compared with those in previous reports. The authors should show Venn diagram for HSF1 peaks between two experiments (in both NHS and HS MCF-7 cells) in Supplementary Figure. Please indicate the number of identified HSF1 peaks in NHS and HS MCF-7 cells in the text. Is HSF1 binding intensity from one experiment used for generation of heatmap in Figure 1c?

In response to the reviewer’s query about Figure 1, in which we had originally shown one dataset, we now include three including a new NHS/HS replicate we generated. As requested, we also add supplementary figure panels for Venn diagrams with NHS and HS peak numbers for replicates in the same condition (Supplementary Fig. 1b) and NHS/HS comparisons per replicate (Supplementary Fig. 6a), and indicate the numbers of NHS and HS peaks in Results section.

Because peak numbers is an important issue, we dug into it some more. Our numbers for MCF7 cells are on the higher side, but they do not exceed what is reported in 52 HSF1 datasets in a transcription factor ChIP database (Czipa et al., 2020) (Figure R2). Regarding the differences in peak numbers for the same cell line, because the same antibodies were used in all studies, antibodies is not it. We thus looked at ChIP noise and sequencing depth as contributing factors. Previous HSF1 HS datasets (Mendillo et al., 2012, Vihervaara et al., 2013) show enrichment in the same regions as our data although they are not callable as peaks (Figure R3). Despite limited data, there is visible cell type specificity – and reproducibility - of HSF1 patterns across studies over the years (Figure R3), even though it can not be picked up bioinformatically through peak calling. In our NHS replicates, peak numbers span the range across multiple cell lines in published replicates. Individual replicates show enrichment visually even if peaks are not called there, so they are more alike even if peak numbers are different (Supplementary

Figure R2, Numbers of transcription factor peaks in public datasets showing numbers of peaks in the public database, left (Czipa et al., 2020) including 52 HSF1 datasets and 3727 if all transcription factor datasets. The database does not distinguish basal from induced cells. Our datasets (GD) our datasets are indicated for each cell line, at NHS and HSR (HS+As).

Fig. 6e, f). We now cite the database and the four new supplementary figure panels mentioned above.

Figure R3. HSF1 HS ChIP-seq tracks for relevant datasets, including public, hESCs (Lyu et al., 2018), K562 (Vihervaara et al., 2013, (K 562 (V)), MCF10A (Mendillo et al., 2012, MCF10A (M)), and our datasets for MCF7 (GD) and K562 (GD). Random regions were chosen with several adjacent enrichment regions across cell lines, with no preference for specific genes. MCF10A and MCF7 Rep 4 and Rep 5 were done with SCBT-9144 antibody (discontinued) and all other tracks with Enzo ADI-SPA-901-D antibody. MCF10A datasets show higher background, possibly because of lower abundance of active HSF1. Datasets are autoscaled individually to the highest peak and are normalized to CPM, except K562 (V) tracks that are obtained from the .tgf file provided in the GEO record. A. Example of cell type selectivity of HSF1 binding locations. All tracks are HS and are CPM-normalized except K562V, which was loaded from the tgf file and has a differently defined Y-axis. B. Individual replicates where available. Asterisks represent cell line-selective peak locations.

To address peak numbers due to sequencing depth, we generated an N=3 MCF7 NHS/HS biological replicate that we sequenced to >130M+ raw read pairs per sample. Progressive subsampling of this high-coverage dataset shows that the numbers of called peaks continue to increase without obvious saturating even beyond ~130M reads, over 4-fold over our normal

sequencing depth, Figure R4A). This, of course, presents a general problem of how far we can go to trust the increasing ChIP-seq peak numbers. The newly called peaks continue to show high enrichment of the HRE sequence motif (Figure R4B) way past the depth of our datasets. So while we do not know when one should stop finding peaks, we had stopped conservatively. We did not scout bioinformatics extensively, although our conclusions do not change with higher stringency of peak definitions (Supplementary Fig. 2e). Peaks arising from higher coverage may reflect the fine details of the nucleosome architecture.

Figure R4. Properties of HSF1 peaks as function of sequencing coverage. The numbers of called peaks are shown versus percentile of sequencing for MCF7 HS sample sequenced to high depth (100% corresponds to ~150M reads). Red dots correspond to coverage that most closely represent the one used in this manuscript for all samples. B. Percentage of peaks containing HRE motif (HRE 1 and HRE 2 as in figure as function of HSF1 peak percentile: 100% in the X axis in B corresponds to the number of peaks in A).

4) The authors stated “An active promoter histone mark H3K4me3 showed no drastic changes in HS compared to untreated control (non-heat shock, NHS) cells (Fig. 1a, d)” (P4). Is this data consistent with previous reports (H3K4me3 level in a specific HSP gene and genome-wide H3K4me3 analysis)?

*No H3K4Me3 increase in HS is consistent with previous studies noting overall histone removal on HSP genes in *Sacharomyces* (Zhao et al, 2005; Zanton and Pugh, 2006), loss of pan H3 upon HS in *Drosophila* cells (Petesch and Lis, 2008), and H2Az-driven nucleosome removal upon gene activation (review by Workman et al., 2006). In addition to the above-mentioned references that we had cited, a study by Li et al., 2017, notes reduction in H3K4Me3 signal on HSP70 in *Drosophila* upon HS. This may be indirectly related to replacement of nucleosomes with those containing H3.3 (Schwartz and Ahmad, 2005, PMID: PMC1074318). Finally, the Corces group (Lyu et al., 2018) published a ChIP-seq dataset that can be used to see the same localized reduction of H3K4Me3 signal in human ESCs as we observe in MCF7 cells. We now*

use Lyu et al data tracks provided under the GEO accession number GSE105028 to update a Supplementary figure (Supplementary Fig. 1f), add the citation and mention it in the text. We did not discuss it here, but at least one study reported an increase in H3K4Me3 upon gene activation with EGF (Edmunds et al., 2008, PMID: PMC2168397). We do not know why that is, but this may reflect different steps of transcription being upregulated, Pol II recruitment versus pause release. We now cite the above references in the manuscript except those with listed PMCID. Also, in response to Reviewer #3, we added a supplementary figure panel showing Pol II and H3K4Me3 signal from our data on activated versus not activated genes wherein Pol II, but not H3K4Me3 promoter signal follows gene activation.

5) The authors stated “Elevated temperature is therefore not a prerequisite for genome-wide HSF1 binding” (P6). The heading of Fig. 2 “Fig. 2: Widespread HSF1 binding is independent of elevated temperature.” (P37). Is it better to state that widespread HSF1 binding is induced by As treatment?

We agree and changed Fig 2 caption accordingly.

6) What “As-exclusive” and “As-specific” mean (P7, P8)?

We used them interchangeably for peaks found in one treatment, but not found in the other, which we agree is confusing. Wording made consistent.

7) In line 168 (P6), “within a cell line (Fig. 4b; Fig. 2f).” Figure 4c should be cited here.

Done

8) Figure 2a and 2b; amend color of boxes.

Done

9) Figure 4b; gene names and structures are too small to see.

We removed gene tracks from the figure and included the genomic coordinates of the region in figure captions so that this region could be readily located online if needed.

10) Figure S6; it is better to add heading of this figure.

We address this query as part of the response to Reviewer #3, for which we consolidated supplementary figures into one supplementary figure per main figure. We added caption to this panel (new Supplementary Figure S2B).

Reviewer #2 (Remarks to the Author):

This study profiles HSF1 binding and gene expression under heat shock response in two conditions and two cell lines. The authors found that HSR activation is marked by widespread elevation of HSF1 binding. HSF1 binding locations are

shared between HSR induced through temperature and arsenic stimuli, but activated genes are stimulus-specific. HSF1 binding is enriched at activated genes, but most HSF1 binding does not activate expression. HSR induced HSF1 binding and gene activation are distinct between MCF7 and K562.

Major limitations:

1. The motivation of this study was not justified in the Introduction section. As molecular profiling of heat shock response has been performed in multiple cell lines and conditions (for example, Mahat et al. 2016, Vihervaara et al. 2017), the significance of this study remains unclear. Results in Fig1 (including elevated HSF1 binding in HSR) and Fig 3 (decoupling between HSF1 binding and gene activation) can be found from multiple similar studies.

We agree we could have explained ourselves better. The reviewer is correct in that HS-dependent increase in HSF1 signal is well-known, and decoupling between HSF1 binding and transcription has been shown from studies including Mahat et al. 2016, and Vihervaara et al., 2017. These studies inspired our work and we now mention this in Introduction and Discussion as also requested by Reviewer #1. There are at least two directions that remained unexplored. The first direction is dynamics of HSF1 patterns. The second is conservation of a connection between HSF1 binding and transcription. Studies by Mendillo and Sistonon groups explored HSR in terms of two transcription factors, HSF1 and HSF2. Our work takes a complementary approach by pivoting around one transcription factor between cell lines and conditions. To the best of our knowledge, such comparisons have not been made for a transcription factor in any system. We updated the Introduction and Discussion sections to clarify this.

Our technical motivation was sparsity of existing HSF1 human HS datasets, detailed in response to Reviewer 1 (item 2). Regarding specific figures, the novelty of Figure 3 is in showing that transcription is more flexible than HSF1 binding. For Figure 1, it sets the flow of the manuscript by narrowing down the focus from multiple possible readouts to HSF1. As pointed by Reviewer #1, the figure addresses a question about the numbers of HSF1 enriched regions, so we feel it is important to the flow of the manuscript.

2. Mechanistic and functional analysis are largely missing from the current manuscript. Several interesting questions arise from the data but the authors didn't attempt to address them. For example, why do arsenic stimuli activate a distinct set of genes from high temperature? Why do increased HSF1 binding activate some genes but not the other? Gene ontology can be applied to uncover gene programs underlying treatments and cell type specific HSR. Motif enrichment analysis can be performed to reveal co-factors of HSF1 to better understand regulatory mechanisms. Unfortunately, such in-depth analysis can rarely be found, as current analysis focuses on describing the data rather than interpreting it. New hypotheses were not raised from the data or tested either.

We thank the reviewer for this point. Answers to these questions have likely to do with co-factors and addressing them is in our long-term vision that drives our ongoing work. The current manuscript was designed deliberately to pivot around a single transcription factor. The PhD student who led it had surmised, and I agreed, that this point of view would be more original than choosing to pursue a pair of factors, which had already been reported in studies by Corces, Lindquist, Lis, Mendillo, and Sistonen groups. One known HSF1 co-factor with a good ChIP antibody is HSF2. While HSF2 physically interacts with HSF1 and has the same apparent DNA sequence preferences, it is still unclear whether HSF1 and HSF2 must co-bind to the same sites on DNA or not. Outside of HSF2, profiling of many transcription factors even with advanced methods such as CUT'N'RUN/tag remains a formidable challenge across the field. This was our technical reason why in this initial study we focused on HSF1, profiling of which we had carefully optimized already. For the sequence context at HSF1 binding sites, we noted hits to a non-HRE motif belonging to Fra1/2 and Fosl2, constituents of the AP-1 complex (Supplementary Fig. 2c). This was seen for HSF1 binding events associated with HS, but not As. AP-1 is widely involved in signal responses and has been noted in relation to heat shock in human ESCs (Lyu et al., 2018). One question evoked by our work is whether a co-factor with the same binding specificity such as HSF2 marks a more conserved or functional portion of HSF1 binding events. Another arises from possible involvement of AP-1 components in a portion of HSF1 binding during HSR, and has to do with the hierarchy in the binding of co-factors and/or DNA- versus co-factor driven TF binding. Both questions were made possible with this work. To clarify these points, we updated the discussion.

Reviewer #3 (Remarks to the Author):

In the current manuscript, Dastidar and colleagues attempted to understand the relationship between HSF1 binding genome-wide and the consequence of gene expression in two different carcinoma cell lines in response to two different external stimuli, temperature elevation, and toxic arsenite supplementation. To investigate the relationship, the authors performed multi-omics, including ChIP-seq, PRO-seq, and ATAC-seq.

The authors designed and executed their planned experiments systematically, and the manuscript is written well. However, this study does not add ample information to the existing finding except for a head-to-head comparison of genome-wide HSF1 binding and differential gene expression between two cell types in response to two different stimuli. Moreover, this study lacks identifying potential molecular mechanisms for cell type- and the stimuli-specific connection between HSF1 binding to DNA and differential gene expression. Therefore, this reviewer thinks this manuscript, in its current content and format, has limited scope in advancing the respective field.

However, this reviewer recommends addressing some concerns to improve the quality of the manuscript in a future submission.

1. Determine and compare side-by-side HSF1 occupancy at NHS and HS conditions for the promoters of genes upregulated explicitly upon heat shock or arsenic treatment with the genes, the expression of which is unchanged.

Overall, activated genes show higher HSF1 signal in HSR, consistent with association of HSF1 binding with activation (Fig. 3b, d). However, on genes that are activated explicitly in one treatment, we see no significant differences in HSF1 binding between HS and As datasets, that is, between activated and nonactivated gene states during HSR (Fig. 3i, k). This indicates that the binding of HSF1 takes place independently of stimulus-specific co-factors, but the function may require them. We updated the wording in discussion.

2. Upon heat shock in MCF7 cells by PRO-seq, the authors identified the upregulation of 2,228 genes. However, I did not see anywhere in the manuscript that the authors analyzed how many HS and non-HS genes. Similarly, the same pieces of information are missing for As-induction. Moreover, the authors did not mention how many genes are upregulated upon As-treatment and how many genes are downregulated upon heat shock and As-treatment.

The numbers of downregulated genes were 1,492 in HS and 3,084 in As, of which 65 (4.4%) and 365 (11.8%) show HSF1 peaks for HS and As, respectively (compared to 28% and 18% of genes activated in HS and As). We did not include repression in an HSF1-centric manuscript because repression is not related to HSF1 binding, as was also shown previously (Mahat et al., 2016), and we investigate it with respect to Pol II pausing in a separate study. For activated genes, we now include their numbers in the main text.

Regarding HS genes, which we believe may refer to known HSP genes as, for example, done in (Vihervaara et al 2013), we had not done that. We thank the reviewer for this question. In response, we added a panel about HSP genes (Supplementary Fig. 3a and 3b, respectively, for MCF7 and K562 cells). We see a close overlap in our data for HSF1 binding and activation between conditions and cell lines in our hands. Like Mahat et al 2016 and Vihervaara et al., 2013, we find about half of HSP genes upregulated in HS, with all HSP genes that were shown as upregulated in Vihervaara et al 2013 upregulated in our list. While the numbers of activated HSP genes are too small for statistics, unlike HS, activation of HSP genes exclusive to As appears not to be connected to HSF1. We find no repurposing of HSF1 to metal stress-specific activation with a possible exception of HSPA14. Overall, known HS-responsive HSP genes work as a conserved cohort activated in concert in HSR induced with either stimulus. We added the respective text to Results section.

3. Line 114-117, Fig 2h, and Supplementary Fig 8: It is unclear whether the authors compare all activated genes or HSF1-bound activated genes between two

conditions (HS and As-treatment). If these figures represent the comparison between all activated genes, then the authors should compare HSF1-occupied activated genes between the two treatments.

All activated genes are compared here. HSF1-bound. Figure legend was updated.

4. Line 117-119: In comment 3, if authors compared only HSF-bound activated genes, then the most relevant conclusion would be "altogether the results suggest depending on the type of upstream signals HSF1 activates different sets of genes in same cell type".

What the reviewer suggests is a stronger statement. We would like to be more cautious here as we do not know whether the signals are in fact upstream or downstream/independent of HSF1 binding. We rephrased this sentence for clarity.

5. Line 126-128 and Fig 3b,c: The authors observed only 1/4th and 1/5th of activated genes' promoters are HSF1-bound in HS and As-treatment, respectively. The authors also identified approximately 1/3rd, and 1/4th of not activated genes' promoters are HSF1-occupied. How about the correlation between HSF1-unbound activated genes and HSF1-bound not activated genes in HS and As-treatments?

We've added the overlaps between these gene categories as a supplementary figure panel (Supplementary Fig. 3c, 3d). Based on these two stimuli, genes with HSF1 peaks appear more conserved than genes without HSF1 peaks, but among HSF1-bound genes, activated genes show less overlap. Apart from denoting a combinatorial nature of regulation by transcription factor, this indicates that the number of functional HSF1 partners may be limited in a given cell type. We thank the reviewer for this point.

6. To understand the relation between HSF1 binding and chromatin accessibility, authors compared ATAC-seq data with HSF1-peaks in NHS and HS conditions. This reviewer wonders whether the chromatin accessibility data correlates well with the gene activation.

We noted an increase in ATAC signal among activated genes (Supplementary Fig. 7b, 7c) compared to nonactivated genes. The increase in ATAC signal is connected to gene activation regardless of whether HSF1 binds or not. The increase in ATAC signal on activated genes is mirrored by an increase in Pol II signal, but not H3K4Me3 as we also highlight in response to Reviewer #1. We found no difference in basal ATAC signal between activated and nonactivated genes, indicating that more open DNA does not necessarily "poise" genes for activation. We added the supplemental figure that pertains to the following comment and updated the text.

7. Sometimes the detailed description of the analysis challenges understanding and extracting the study's big picture.

We have gone through Results and removed some points to focus less on details while incorporating suggested analyses. In particular, we deleted mutations analyses and moved scatterplots, description of nonspecific “tusk” analysis that validates normalization, and moved quartile switching to supplementary material. We also streamlined the Results section about ATAC and edited other results sections.

8. The authors can consider combining the supplementary figures into fewer.

Done, by consolidating individual figures, now reflecting one supplementary figure per main figure, and deleting some supplementary figures altogether. Thank you for the suggestion.